# Hydrogen sulfide stimulates *Mycobacterium tuberculosis* respiration, growth and pathogenesis

Vikram Saini [1,2,8], Krishna C. Chinta [1], Vineel P. Reddy [1], Joel N. Glasgow [1], Asaf Stein[3], Dirk A. Lamprecht [4,9], Md. Aejazur Rahman [4], Jared S. Mackenzie[4], Barry E. Truebody [4], John H. Adamson[4], Tafara T.R. Kunota [4], Shannon M. Bailey [3], Douglas R. Moellering [2,5], Jack R. Lancaster Jr.[6] & Adrie J.C. Steyn [1,2,4,7]*

Hydrogen sulfide ($H_2S$) is involved in numerous pathophysiological processes and shares overlapping functions with CO and ●NO. However, the importance of host-derived $H_2S$ in microbial pathogenesis is unknown. Here we show that *Mtb*-infected mice deficient in the $H_2S$-producing enzyme cystathionine β-synthase (CBS) survive longer with reduced organ burden, and that pharmacological inhibition of CBS reduces *Mtb* bacillary load in mice. High-resolution respirometry, transcriptomics and mass spectrometry establish that $H_2S$ stimulates *Mtb* respiration and bioenergetics predominantly via cytochrome *bd* oxidase, and that $H_2S$ reverses ●NO-mediated inhibition of *Mtb* respiration. Further, exposure of *Mtb* to $H_2S$ regulates genes involved in sulfur and copper metabolism and the Dos regulon. Our results indicate that *Mtb* exploits host-derived $H_2S$ to promote growth and disease, and suggest that host-directed therapies targeting $H_2S$ production may be potentially useful for the management of tuberculosis and other microbial infections.

[1] Department of Microbiology, University of Alabama at Birmingham, Birmingham, AL, USA. [2] Center for Free Radical Biology, University of Alabama at Birmingham, Birmingham, AL, USA. [3] Department of Environment Health Sciences, University of Alabama at Birmingham, Birmingham, AL, USA. [4] Africa Health Research Institute, Durban, South Africa. [5] Department of Nutrition Sciences, University of Alabama at Birmingham, Birmingham, AL, USA. [6] Departments of Pharmacology and Chemical Biology, Medicine, and Surgery, University of Pittsburgh School of Medicine, Pittsburgh, PA, USA. [7] Center for AIDS Research, University of Alabama at Birmingham, Birmingham, AL, USA. [8] Present address: Laboratory of Infection Biology and Translational Research, Department of Biotechnology, All India Institute of Medical Sciences, New Delhi, India. [9] Present address: Janssen Infectious Diseases and Vaccines, Janssen Pharmaceutica NV, Beerse, Belgium. *email: asteyn@uab.edu

**M**ycobacterium tuberculosis (Mtb), the etiological agent of TB, can persist in a clinically latent disease state for decades. Identifying host factors that influence Mtb survival is essential for understanding how Mtb persists. Physiologic gaseous messengers such as carbon monoxide (CO) and nitric oxide (•NO) are important to the outcome of Mtb infection. In particular, •NO produced by inducible nitric oxide synthase (iNOS) exerts a bacteriostatic effect on Mtb by inhibiting respiration[1] and is crucial for protection against TB[2,3]. Notably, •NO[1], CO[4] and low oxygen (O$_2$) levels[5] induce the expression of the 48-gene Dos dormancy regulon, which is associated with reduced respiration and bacterial replication. Hydrogen sulfide (H$_2$S) is an emerging signaling molecule that is produced enzymatically in mammals by cystathionine β-synthase (CBS), cystathionine γ-lyase (CSE) and 3-mercaptopyruvate sulfurtransferase (3-MST)[6,7]. At physiological pH, H$_2$S exists mainly as a hydrosulfide anion (HS$^-$, 75−80%), with the remainder in the form of the dissolved uncharged H$_2$S gas (~20−25%) and negligible amounts of S$^{2-}$[8]. Here, unless specified, we refer to the sulfide pool (H$_2$S + HS$^-$ + S$^{2-}$) as H$_2$S.

H$_2$S modulates various physiological functions including inflammation, metabolism, neuroprotection, vasodilation[9], and protection of microbes against antibiotics[10,11]. Surprisingly, despite having several overlapping functions with CO and •NO[8,12], a role for host-generated H$_2$S in bacterial pathogenesis has not yet been described. H$_2$S can modulate mitochondrial respiration resulting in a suspended animation-like state in mice[13]. In mammals, H$_2$S elicits a biphasic, concentration-dependent mitochondrial response[14], which can be cytotoxic or cytoprotective. For example, at high concentrations H$_2$S reversibly inhibits cytochrome c oxidase (Complex IV)[15–17]. In contrast, at low concentrations H$_2$S can serve as bioenergetic fuel to stimulate mitochondrial respiration without uncoupling of respiration[16,18]. Due to the complex chemistry of H$_2$S and lack of routine assays for accurate measurement, the physiological concentration of H$_2$S in vivo has been a matter of debate[19–21]. Here we hypothesize that host-derived H$_2$S plays a role in TB through modulation of Mtb respiration and bioenergetics. To test this hypothesis, we performed Mtb infection studies in CBS-deficient (Cbs$^{+/-}$) mice, which have reduced H$_2$S levels. We also examined the role of CBS in inflammation, determined the total sulfide levels in infected mice, and examined how H$_2$S modulates Mtb growth, gene expression, metabolism and respiration.

## Results

**Mtb infection alters CBS levels in macrophages.** Infection of RAW 264.7 macrophages with Mtb H37Rv resulted in a 34-fold increase in the level of CBS (Fig. 1a, b), a major enzymatic source of H$_2$S in mammalian cells[22,23]. This suggested that increased H$_2$S production is a host response to Mtb infection. To investigate the role of H$_2$S in Mtb virulence and pathogenicity, we used mice deficient in H$_2$S production due to disruption of Cbs, the gene encoding CBS[22,23]. Since the Cbs$^{-/-}$ genotype is perinatally lethal, we used heterozygous (Cbs$^{+/-}$) mice that have a normal life span and show an ~50% reduction in CBS levels[22]. Infection of Cbs$^{+/+}$ (WT) peritoneal macrophages resulted in a 25-fold increase in the level of CBS, whereas the levels of CSE and 3-MST did not change appreciably (Fig. 1c, d). Bacterial recovery from Mtb-infected Cbs$^{+/-}$ peritoneal macrophages was reduced compared to WT cells (Fig. 1e), which correlates with lower H$_2$S levels in these cells as measured by an H$_2$S-specific probe (Fig. 1f). Mtb recovery from Cbs$^{+/-}$ macrophages treated with the H$_2$S donor GYY4137[24] was increased compared to Cbs$^{-/+}$ cells alone (Fig. 1e). GYY4137 slowly releases H$_2$S thereby mimicking physiological release. This result suggests that differences in Mtb recovery are due primarily

to H$_2$S. Lastly, we observed no impact on bacterial recovery from infected peritoneal macrophages treated with homocysteine (10, 25 or 50 μM), ruling out mild hyperhomocysteinemia observed in Cbs$^{+/-}$ mice (<15 μM)[22,25] as a contributor to reduced bacterial burden (Supplementary Fig. 1).

**Cbs$^{+/-}$ mice survive longer following Mtb infection.** The macrophage infection studies above clearly show decreased bacillary growth in H$_2$S-deficient Cbs$^{+/-}$ cells and that Mtb growth could be chemically complemented with H$_2$S. To more clearly establish the role of H$_2$S in Mtb virulence and pathogenicity, we infected WT and Cbs$^{+/-}$ mice with Mtb and observed that Cbs$^{+/-}$ mice survived significantly longer than WT mice (median time to death 286 days vs. 211 days for WT mice (Fig. 1g)). CBS levels in the lungs of uninfected Cbs$^{+/-}$ mice were significantly lower than in WT, and CBS levels in the lungs of WT mice increased significantly at 8 weeks post infection (Fig. 1h, i).

At 8 and 14 weeks post infection, Cbs$^{+/-}$ mice had ~6-fold and ~4-fold fewer Mtb bacilli in the lungs, respectively, compared to WT mice (Fig. 1j). Similarly, we observed ~4.5-fold and ~2.5-fold fewer bacilli in spleens of Cbs$^{+/-}$ mice compared to WT mice at 8 and 14 weeks post infection, respectively (Fig. 1k). While significant, these modest differences in organ burden contrast with the striking difference in survival (Fig. 1g). Furthermore, gross pathology (Fig. 1l) and histopathology appraisal (Fig. 1m, n) of infected WT and Cbs$^{+/-}$ mice were consistent with the observed organ burden. We noted moderate to large coalescence of multiple granulomatous lesions with infiltration of lymphocytes in the lungs of WT mice whereas the lung architecture in Cbs$^{+/-}$ mice was relatively intact with fewer small granulomatous lesions (Fig. 1m). In sum, our data suggest that Mtb infection of host cells upregulates CBS production, leading to increased H$_2$S levels, which exacerbate disease as indicated by reduced host survival, and to a lesser extent by increased organ burden. These findings identified a new phenotypic category of host genes that exacerbates TB disease whereas their disruption prolongs survival during infection.

**Immune responses in Mtb-infected mice.** To determine whether reduced bacillary burden and increased survival of Cbs$^{+/-}$ mice could be attributed to immunological differences between Cbs$^{+/+}$ and Cbs$^{+/-}$ mice, we examined host immune responses in Cbs$^{+/-}$ and WT mice pre- and post-Mtb infection. In uninfected mice (Supplementary Figs. 2A−J, 3B, C), except in the spleens of Cbs$^{+/-}$ mice which showed fewer macrophages (Supplementary Fig. 2F), we observed no significant differences in the percentages of myeloid or lymphoid cell types in the lungs and spleens of Cbs$^{+/-}$ and WT mice, even though the levels of proinflammatory IL-1α were elevated and anti-inflammatory IL-10 were reduced in Cbs$^{+/-}$ compared to WT mice (Supplementary Fig. 3A). To distinguish whether the reduced inflammatory response in Cbs$^{+/-}$ mice during chronic infection is due to lower CBS protein levels or reduced bacillary burden, we analyzed the immune response at an early time after infection (3 weeks) when the lung bacillary burden was similar (Supplementary Fig. 4A), but serum sulfide levels differed between WT, and Cbs$^{+/-}$ mice (Supplementary Fig. 4B). As shown in Supplementary Figs. 2K-R and 3D and E, except for increased percentages of splenic CD4$^+$ T cells (Supplementary Fig. 2P), reduced percentages of CD4$^+$IFN-γ$^+$ T cells (Supplementary Fig. 2R) and reduced percentages of CD11b$^+$Gr-1$^-$ myeloid cell in the spleens (Supplementary Fig. 3E) of Cbs$^{+/-}$ mice, there were no significant differences in percentages of myeloid or CD4$^+$ T cells in the lungs and spleens of Cbs$^{+/-}$ and WT mice at 3 weeks post infection. During the chronic stage of infection (8 weeks), cytokine levels in Cbs$^{+/-}$ and

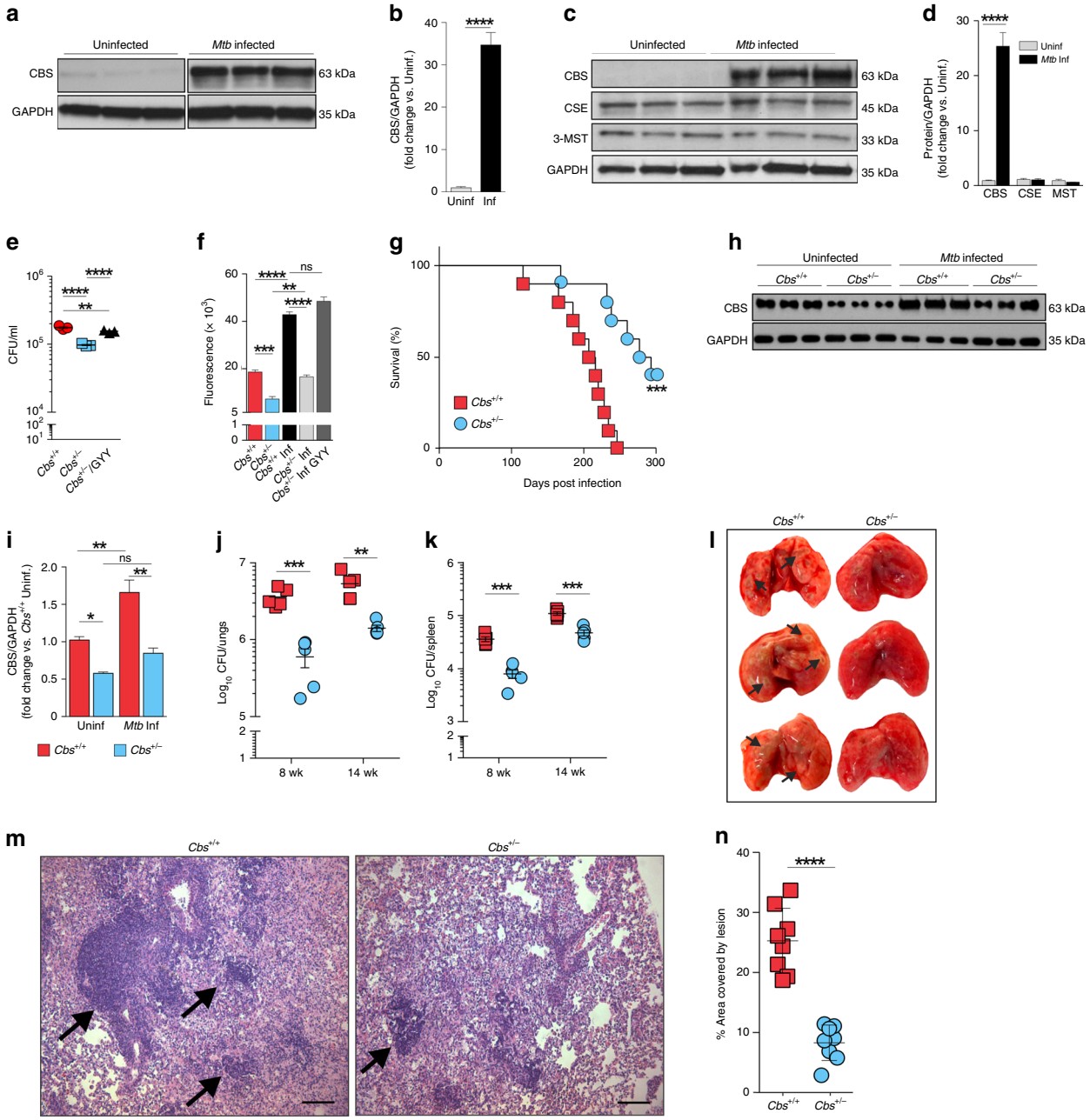

**Fig. 1 Attenuation of *Mtb* virulence and pathogenicity in *Cbs*$^{+/−}$ mice. a** Representative western blot and **b** densitometry analysis of CBS levels in RAW 264.7 macrophages following 6 h of *Mtb* infection ($n = 3$). **c** Representative western blot and **d** densitometry analysis of H$_2$S-producing enzymes in WT mouse peritoneal macrophages following 6 h of *Mtb* infection ($n = 3$ animals per group). **e** CFU recovery from *Mtb*-infected peritoneal macrophages isolated from WT and *Cbs*$^{+/−}$ mice at 36 h post infection. **f** Relative quantitation of H$_2$S production from uninfected or infected (Inf) peritoneal macrophages using the H$_2$S-specific fluorescent probe WSP1[28] at 36 h post infection. The *Cbs*$^{+/−}$ Inf GYY4137 group received 25 μM GYY4137. (**e**) and (**f**) represent pooled data from two or more independent experiments ($n = 4−6$). **g** Survival of mice following *Mtb* infection, representative of two independent survival studies ($n = 10$ animals/group). **h** Representative western blot and **i** densitometry analysis of CBS in the lungs of age-matched mice without infection or 8 weeks post *Mtb*-infection ($n = 3$ animals per group). **j** Bacillary burden in lungs and **k** spleens at 8 and 14 weeks post infection ($n = 5$ animals per group). **l** Representative gross images of infected lungs at 8-week post *Mtb*-infection. Arrows indicate TB granulomatous lesions. **m** H&E staining of representative infected lung sections at 8 weeks post *Mtb*-infection. Arrows indicate area of lymphocyte infiltration (scale bar = 100 μm). **n** Quantitative histochemical appraisal of lesions in infected lungs of WT and *Cbs*$^{+/−}$ mice. Granulomatous region analysis data from three independent animal experiments with 2−3 mice per experiment. Data are shown as mean ± SEM, except (**d**) and (**n**) (mean ± SD). Data (**b**, **d**, **j**, **k** and **n**) were analyzed by unpaired parametric *t* test. Survival data (**g**) were analyzed by Log-rank (Mantel−Cox) test. Data in (**e**) and (**f**) were analyzed by one-way analysis of variance (ANOVA) with Tukey's multiple comparisons test. Data in (**i**) were analyzed by two-way ANOVA with Tukey's multiple comparisons test. *$P < 0.05$, **$P < 0.01$, ***$P < 0.001$ and ****$P < 0.0001$. Source data are provided as a Source Data file.

WT mice were similar except for a lower TNF-α and IL-10 levels in $Cbs^{+/-}$ mice (Supplementary Fig. 3A), which might be due to reduced percentages of neutrophils and lymphoid cells mostly in the spleen (Supplementary Figs. 2S-Z, 3F and G). We observed no clear predominance of either a Th1 or Th2 response, as indicated by the levels of proinflammatory (IL1α, IL-6, IL-12, INF-γ and TNF-α), anti-inflammatory (IL-5, IL-10) and regulatory (IL-17) cytokines in the sera of $Cbs^{+/-}$ and WT mice (Supplementary Fig. 3A). Overall, these data suggest that the differences in bacillary burden and increased survival in $Cbs^{+/-}$ mice are likely not attributable to immunological differences alone, and are more likely due to direct action of $H_2S$ on *Mtb*.

**$H_2S$ modulates mycobacterial growth and energy homeostasis.** In eukaryotes, depending on the concentration, $H_2S$ can either stimulate or inhibit the electron transport chain (ETC) resulting in altered oxidative phosphorylation (OXPHOS) and ATP production[16,26]. Reported physiological levels of $H_2S$ in tissue are controversial because of the assay methods employed and volatility of the gas[20]. However, the current consensus is that nanomolar concentrations of $H_2S$ seem most plausible[19–21]. Unfortunately, this low concentration is below the limit of detection (1 μM) of routine assays such as the methylene blue, 5,5′-dithiobis-2-nitrobenzoic acid and bismuth chloride assays[19,20].

Since our in vivo data suggest that host-generated $H_2S$ interacts with *Mtb* to modulate disease, we tested the hypothesis that $H_2S$ directly modulates *Mtb* growth. For these in vitro studies, we employed GYY4137, a widely used $H_2S$ donor that releases low amounts of $H_2S$ over a sustained period[24,27]. Unlike widely used •NO and CO donors such as diethylenetriamine NONOate (DETA NONOate) and CO-releasing molecules (CO-RMs), respectively, GYY4137 is an unusually slow $H_2S$ releaser wherein the final molar concentration of $H_2S$ released is significantly lower than the starting concentration of GYY4137. These aspects, along with the volatility of the gas[20], make detection of $H_2S$ by standard assays challenging[24,27]. Therefore, we used LC-MS/MS to first confirm spent (decomposed) GYY4137, and the decomposition of GYY4137 in the presence of *Mtb* cells under our conditions into the corresponding molecular ions with $m/z$ of 272.05 $(M-H)^-$, 187.02 $(M-H)^-$ and 88.08 (morpholine $M+H)^+$ (Supplementary Fig. 5), and carefully considered the complex physicochemical properties of $H_2S$ in the design of our experiments (see Materials and methods).

To correlate the amount of $H_2S$ generated in vivo with our in vitro experiments, we used a fluorescent $H_2S$-specific probe, WSP1, to measure levels of $H_2S$ produced over time by macrophages following *Mtb* infection[28]. We then determined which concentrations of GYY4137 generated similar levels of fluorescence with WSP1 over time due to $H_2S$ release. We then exposed *Mtb* cells to this range of GYY4137 concentrations that mimics levels of $H_2S$ produced physiologically by macrophages following infection (Supplementary Fig. 6A, Fig. 1f). As expected, 5, 10, and 25 μM GYY4137 significantly stimulated growth. At day 12, maximum bacterial growth was observed with 25 μM GYY4137 (Fig. 2a, b). In contrast, 50 and 100 μM GYY4137 reduced the growth stimulation in a dose-dependent manner (Fig. 2a, b). Notably, we observed no growth change in *Mtb* control cultures exposed to 25 μM spent GYY4137, comprised of components of the chemical backbone alone (Supplementary Figs. 5 and 6B). These data establish that low levels of sulfide exert a concentration-dependent bimodal effect on *Mtb* growth.

Since enhanced growth is coupled to increased OXPHOS and ATP levels, and low levels of $H_2S$ can stimulate mitochondrial respiration[14,18], we hypothesized that $H_2S$ targets the *Mtb* ETC to modulate energy metabolism. We exposed *Mtb* cultures to

GYY4137 and observed a dose-dependent increase in ATP levels, which was lost in cells exposed to 100 μM GYY4137 (Fig. 2c). Next, we performed extracellular flux analysis, which we have adapted to analyze *Mtb* bioenergetics[29–31], to determine the effect of $H_2S$ on *Mtb* respiration in real time in a noninvasive manner. Again, we observed a bimodal response to GYY4137, wherein low concentrations resulted in a significant dose-dependent increase in the oxygen consumption rate (OCR) and 100 μM GYY4137 mitigated the increased OCR (Fig. 2d, e). Importantly, spent GYY4137 did not alter the basal OCR of *Mtb* cells (Supplementary Fig. 6C). The effects of sulfide on mycobacterial OCR were independently validated using another donor, NaHS (Supplementary Fig. 6D) and by employing an Oroboros O-2k respirometer with the nonvirulent vaccine strain *Mycobacterium bovis* BCG. The bimodal effect of $H_2S$ on growth was also recapitulated with NaHS in *Mycobacterium bovis* BCG (Supplementary Fig. 6E) and *Mtb* (Supplementary Fig. 6F).

In sum, our results demonstrate that low levels of $H_2S$ target the ETC to increase respiration and ATP levels, leading to increased growth. The capacity of $H_2S$ to positively regulate the ETC, a key component of central metabolism, to induce a hypermetabolic state is likely to benefit *Mtb* during early infection.

**$H_2S$ stimulates *Mtb* central metabolism.** Since respiration is tightly linked to central energy metabolism, we investigated the effect of $H_2S$ on *Mtb* metabolism in more detail. We performed LC-MS/MS-based targeted metabolomic analysis of *Mtb* cells at 1 and 24 h post-exposure to GYY4137 and observed increased levels of the glycolytic intermediates glucose-6-phosphate, fructose-1,6-bisphosphate and pyruvate. Increased levels of these metabolites in the presence of excess ATP demonstrates augmented glycolytic flux (Fig. 2f), since this pathway would be otherwise inhibited when excess ATP is present[32]. Levels of the TCA cycle intermediates citrate and isocitrate were also increased, suggesting that feedback inhibition due to elevated NADH levels is not occurring (Fig. 2f). TCA cycle intermediates such as citrate also augment fatty acid biosynthesis to support mycobacterial growth[32]. Increased NADH, which feeds into the ETC to support increased ATP production is consistent with the ability of $H_2S$ to stimulate growth (Fig. 2a, b) and respiration (Fig. 2d, e). It should be noted that small changes in absolute levels of metabolite pools can have critical implications for metabolism, as this points to dysregulation of metabolic balance that must be maintained at all costs[33,34]. Carbon rearrangement in metabolites is an early indicator of metabolic dysregulation prior to changes in abundance and can be measured using stable isotopes. Therefore, to further examine how $H_2S$ reprograms *Mtb* metabolism, we performed a series of carbon tracing experiments. We first confirmed that the mass distribution vectors (MDV) of glutamate and α-ketoglutarate were identical, thereby indicating that isotopic steady state had been reached under our culture conditions (Supplementary Fig. 7A, B). To determine the fuel source (glucose, acetate, or other endogenous carbon sources (e.g., amino acids)) contributing to carbon fractions in glycolytic and TCA metabolites, we monitored the fate of the uniformly labeled stable isotope $[^{13}C_6]$ glucose and $[^{13}C_2]$ acetate, and calculated the fractional nutrient contribution (FNC; Fig. 3a)[33]. We found that exposure to GYY4137 reroutes carbon away from downstream glycolytic intermediates towards cell wall synthesis and the pentose phosphate pathway (PPP) in a dose-dependent manner. For example, increasing concentrations of GYY4137 increased glucose carbon flow into glucose/fructose-6-phosphate (G/F6P) relative to the spent GYY4137 control and reciprocally reduced carbon flow from other endogenous supplementary carbon

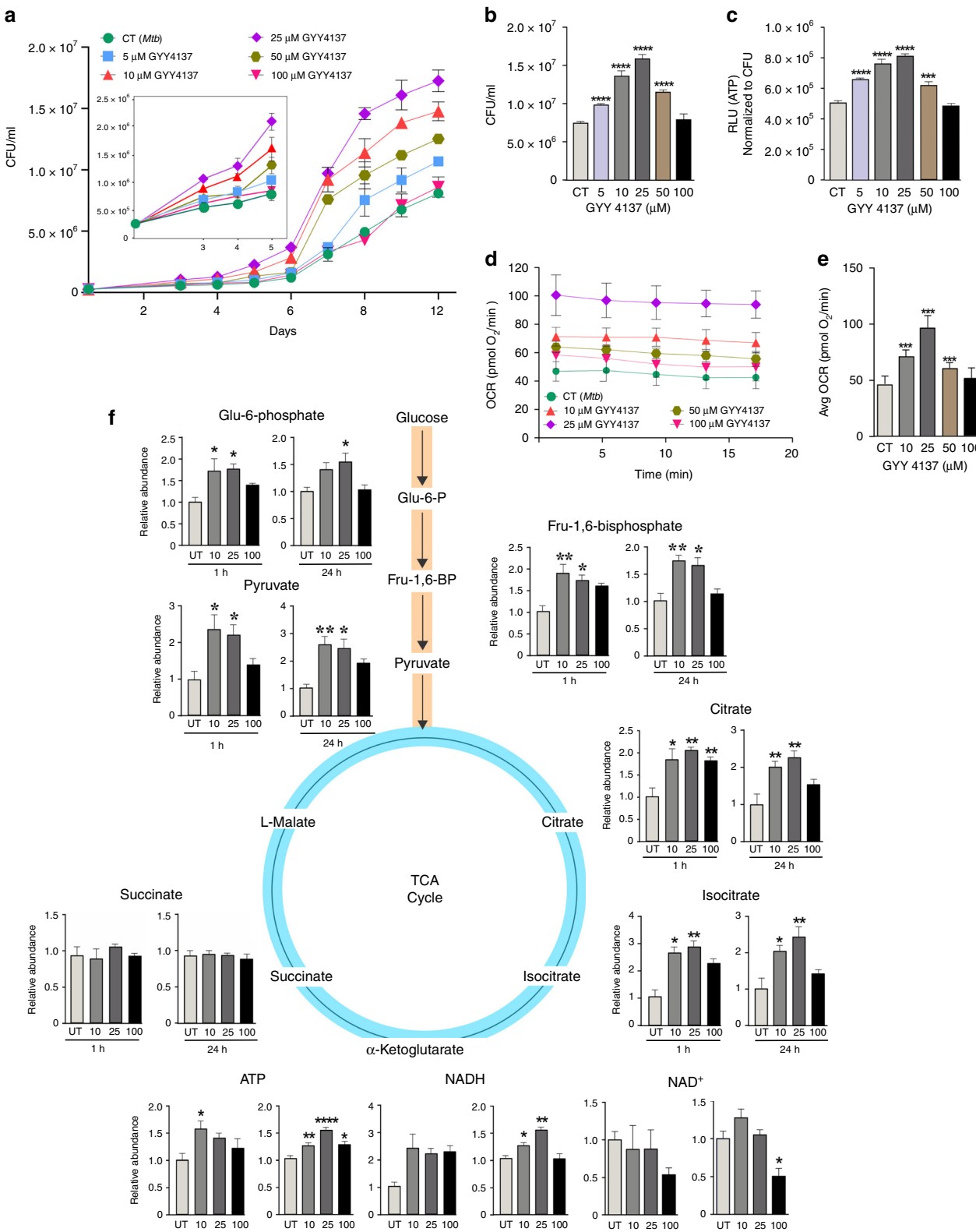

**Fig. 2 GYY4137 modulates *Mtb* growth, bioenergetics and ATP production. a** CFU-based growth profile of *Mtb* following incubation with GYY4137. Inset shows early *Mtb* growth. **b** Mean CFU values at day 12 of the growth analysis. **c** ATP levels in *Mtb* following 24 h exposure to GYY4137. Data (**a**–**c**) are representative of at least two independent experiments. **d** OCR profile ($n = 4$–$6$) and **e** average OCR of *Mtb* exposed to GYY4137 ($n = 20$–$30$). Control (CT) groups in panels (**a**–**e**) received spent GYY4137. **f** LC-MS targeted metabolic analysis of *Mtb* with or without exposure to 10, 25 and 100 μM GYY4137 (for 1 or 24 h). Graphs show relative abundance (peak area) of metabolites in *Mtb* cells ($n \geq 3$ per group). Data are shown as mean ± SD, except (**b**) and (**f**) (mean ± SEM). Data in (**a**) were analyzed using two-way ANOVA with Tukey's multiple comparisons test, with day 12 data shown in (**b**), or one-way ANOVA (**c**–**f**) with Sidak's multiple comparisons test. All statistical differences are relative to untreated controls. *$P < 0.05$, **$P < 0.01$, ***$P < 0.001$, and ****$P < 0.0001$. Source data are provided as a Source Data file.

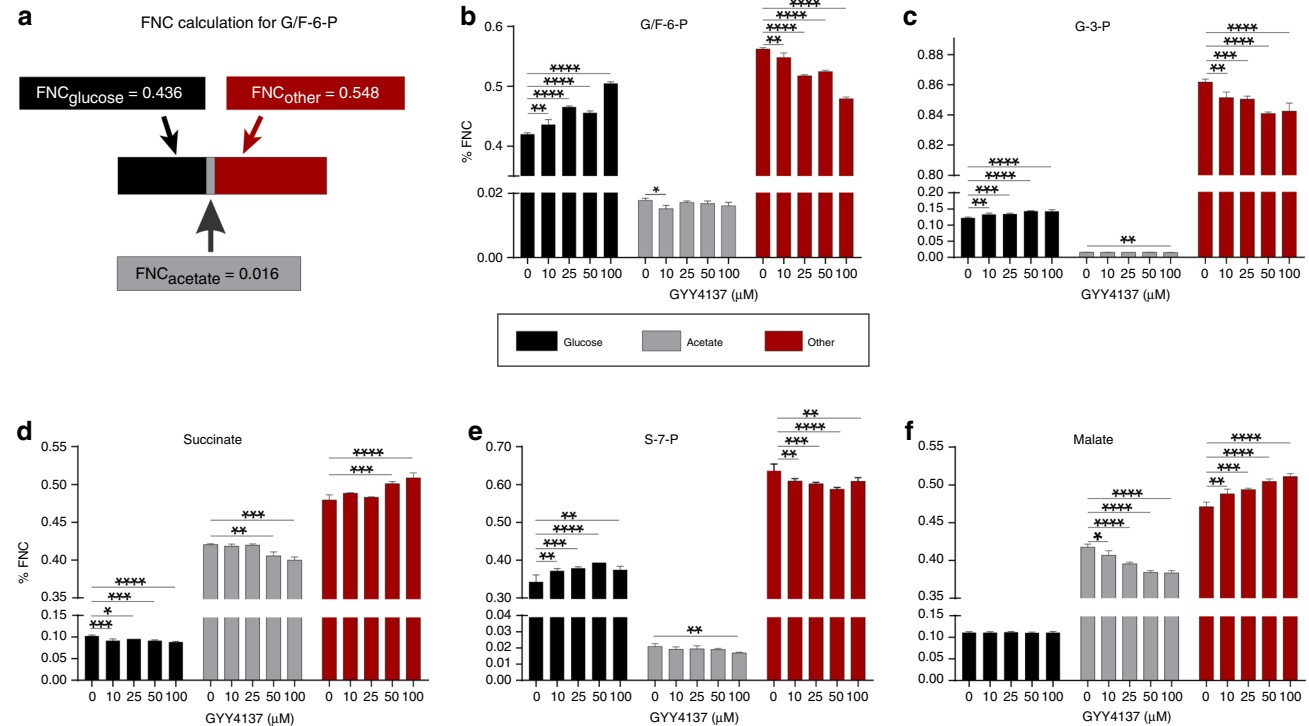

**Fig. 3 Fractional nutrient contribution (FNC) of glucose and acetate exposed to GYY4137. a** FNC was calculated according to a previous study[33]; see also Materials and Methods. **b, c** Glycolysis is stimulated by sulfide as is evident by an increased FNC$_{glucose}$ of G/F6P that corresponds with increased GYY4137 concentration, whereas the FNC$_{acetate}$ and FNC$_{other}$ decreases. The FNC$_{glucose}$ of G-3-P (**c**) and S-7-P (**e**) increases with higher concentrations of GYY4137. **d, f** The utilization of glucose is skewed towards cell wall synthesis and the PPP leading to a decrease in FNC$_{glucose}$ and FNC$_{acetate}$ of succinate and malate as GYY4137 concentration increases, as well as a corresponding increase in FNC$_{other}$ of these two metabolites. Data are representative of two independent experiments ($n = 5$) and are shown as mean ± SD. Data were analyzed using one-way ANOVA with Sidak's multiple comparisons test. All $P$ values are relative to untreated controls (0 μM GYY4137) for individual metabolites. *$P < 0.05$, **$P < 0.01$, ***$P < 0.001$, and ****$P < 0.0001$. Source data are provided as a Source Data file.

sources into this metabolite (Fig. 3b). Secondly, an increase in the FNC of glucose into glycerol-3-phosphate (G3P) and sedoheptulose-7-phosphate (S7P) suggests reprogramming of metabolism towards cell wall biosynthesis and the PPP, which produces NADPH and nucleosides necessary for lipid anabolism and DNA synthesis, respectively (Fig. 3c, e). Notably, the total FNC from other carbon sources (~84%) into G3P was significantly higher than the FNC from glucose (~12%) and acetate (~1%; Fig. 3c). Thirdly, increased levels of GYY4137 reduced entry of glucose or acetate carbons into the TCA cycle intermediate succinate, whereas the FNC of other carbon sources to succinate is increased (Fig. 3d). Interestingly, entry of glucose carbons into malate was unchanged (Fig. 3f), whereas entry of acetate carbons into malate and succinate was reduced with the reciprocal increase from other carbon sources (Fig. 3d, f).

Overall, the carbon tracing data indicate that enhanced energy flux through glycolysis, the PPP and TCA cycle following H$_2$S exposure increases *Mtb* energy metabolism and anabolic substrate intermediates for growth. Lastly, the FNC data demonstrate that H$_2$S rewires energy flux through these pathways by modulating the fraction of the metabolite's carbon produced from the respective fuel source.

**H$_2$S exerts it bioenergetic effect predominantly through CytBD.** We next examined how H$_2$S stimulates *Mtb* respiration and growth. Mycobacteria possess two terminal respiratory oxidases that provide substantial respiratory flexibility as electrons from the menaquinone pool can be channeled either to an *aa$_3$*-type cytochrome c oxidase via the cytochrome *bc$_1$* (cytochrome

*bc$_1$/aa$_3$*) complex or directly to cytochrome *bd*-type menaquinol oxidase (CytBD)[35]. CytBD can partially compensate for a loss in activity of the *bc$_1$-aa$_3$* complex[36], and is required during the transition from acute to chronic *Mtb* infection in mouse lungs to promote bacterial virulence[37]. Potassium cyanide (KCN) inhibits the *aa$_3$*-type cytochrome c oxidase, but not CytBD in *Mycobacterium smegmatis* (*Msm*)[35]. Also, CytBD in *E. coli* has been shown to be less sensitive to cyanide and sulfide than *aa$_3$*-type cytochrome c oxidase[38]. To determine whether H$_2$S exerts its effect through CytBD or the *bc$_1$-aa$_3$* complex, we exposed *Msm* cells to KCN and/or NaHS in an Oroboros O-2k respirometer. We observed that addition of NaHS to *Msm* leads to an increase in basal OCR (Fig. 4a), consistent with the effects of H$_2$S on *Mtb*. In contrast, addition of KCN resulted in a steady decline in the basal OCR of *Msm* cells. Interestingly, addition of NaHS to KCN-treated cells triggered an increase in OCR (Fig. 4a). To define the effect of H$_2$S on *Mtb* growth and ATP production in more detail, we used *Mtb rv1620c::aph* (hereafter referred to as *cydC::aph*), which contains the kanamycin resistance gene (*aph*) inserted in *cydC* (*rv1620c*) as described[37]. *Mtb cydC* encodes a membrane protein linked to the incorporation of a heme cofactor into CytBD. *Mtb cydC::aph* is defective in CytBD activity and shares the phenotype of *Mtb cydAB* mutants[30,39]. Importantly, *Mtb cydC::aph* produces only a functional cytochrome *bc$_1$/aa$_3$*[40]. As a control, we used a complemented strain, *Mtb cydC::aph*-Comp, obtained by expressing *cydC* from an integrated plasmid. Effective complementation was demonstrated by inhibiting cytochrome *bc$_1$* using Q203 as described[30] (Supplementary Fig. 8A), which reroutes electrons to CytBD leading to an increase in OCR

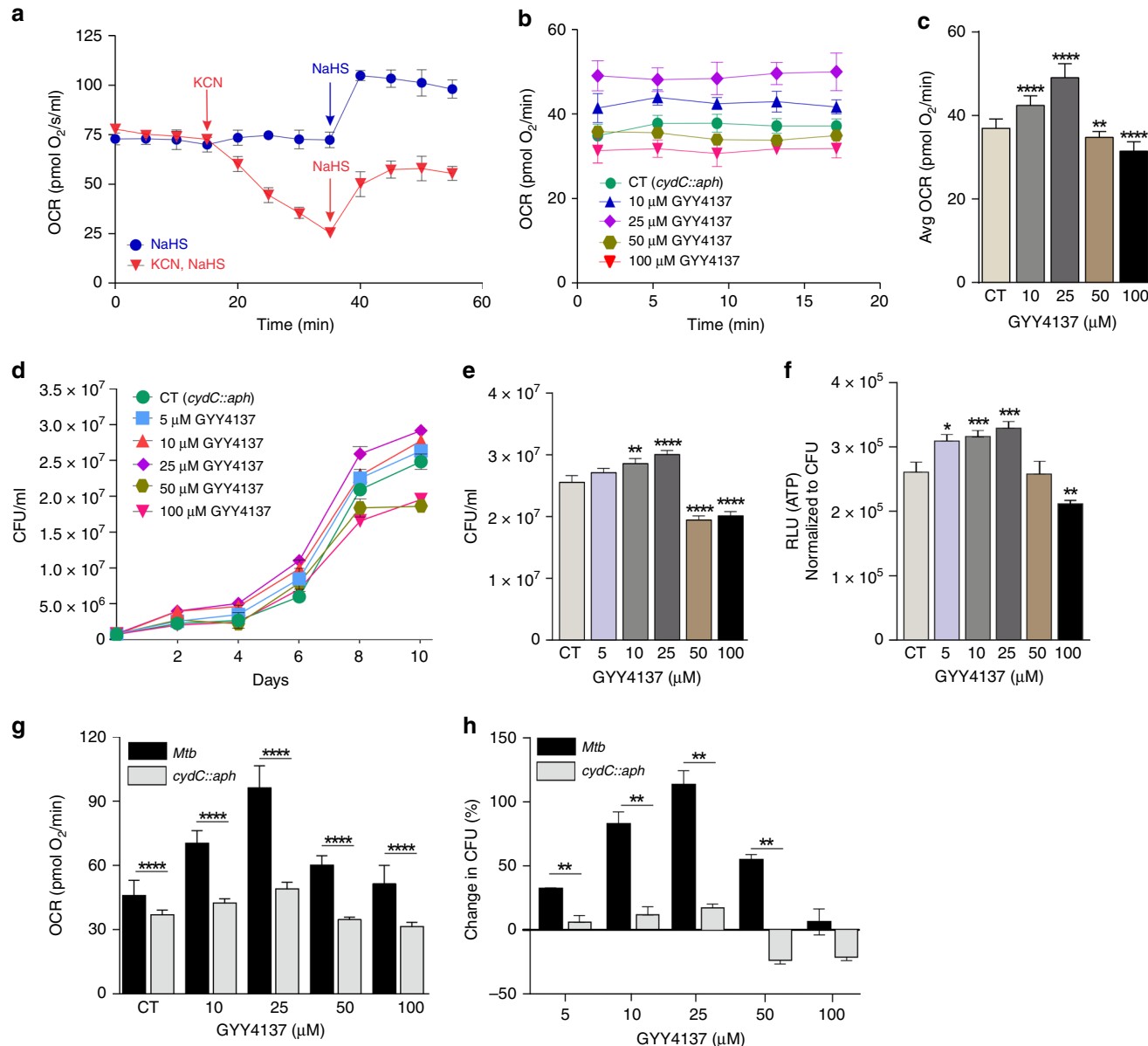

**Fig. 4 H2S stimulates *Mtb* bioenergetics primarily through cytochrome *bd*-type quinol oxidase. a** Representative OCR profile of *M. smegmatis (Msm)* cells. Arrows indicate the time of addition of KCN (100 μM) or NaHS (25 μM) ($n = 5$). **b** Representative OCR profile and **c** average OCR of *cydC::aph* cells exposed to GYY4137 ($n = 12–18$, data pooled from at least two independent experiments). **d** CFU-based growth profile of *cydC::aph* cells following exposure to GYY4137. **e** Mean *cydC::aph* CFU values at day 10 of the growth analysis. **f** ATP levels in *cydC::aph* after 24 h exposure to GYY4137. Data (**d**–**f**) are representative of at least two independent experiments. **g** Comparison of the dose-dependent effect of GYY4137 on the OCR of *Mtb* and *cydC::aph* cells (from Figs. 2e and 4c). **h** Percent change in CFU of *Mtb* and *cydC::aph* (**e**) following exposure to increasing concentrations of GYY4137 (from Figs. 2b and 4e). The percent change was calculated against the CFU of control cells at day 10 following GYY4137 exposure. Data are shown as mean ± SD, except (**e**) (mean ± SEM). All comparisons are against control cells (CT) exposed to spent GYY4137 except in (**g**) and (**h**), which compare *Mtb* and *cydC::aph* at equivalent GYY4137 concentrations. Data were analyzed using two-way ANOVA (**b**, **d**) with Sidak's multiple comparisons test, with average OCR (**c**) or CFU at day 10 (**e**). Other data were analyzed by one-way ANOVA with Sidak's multiple comparisons test (**f**) or by using an unpaired parametric *t* test for comparisons at each GYY4137 concentration (**g** and **h**). *$P < 0.05$, **$P < 0.01$, ***$P < 0.001$, ****$P < 0.0001$. Source data are provided as a Source Data file.

(Supplementary Fig. 8B and C). Complementation was also demonstrated by the marked increase in OCR of WT *Mtb* and *Mtb cydC::aph*-Comp cells compared to *Mtb cydC::aph* in the presence of 25 and 100 μM GYY4137 (Supplementary Fig. 8D and E). Although *E. coli cydD* mutants were shown to be susceptible to high (20 mM) concentrations of cysteine[41], we found that *Mtb cydC::aph* is not susceptible to physiological levels of cysteine (~128 and ~250 μM in human cells and plasma, respectively)[42,43] (Supplementary Fig. 9). Next, we measured the OCR in *cydC::aph* cells following exposure to GYY4137 and observed a

small, but significant increase in OCR at 10 and 25 μM GYY4137 compared to control cells receiving spent donor (Fig. 4b, c). Notably, at higher GYY4137 concentrations (≥50 μM), not only is the stimulatory effect on the OCR lost, but the OCR is repressed below the basal level, indicating an inhibition of cytochrome *bc1/aa3* oxidase in *cydC::aph* cells (Fig. 4b, c). *CydC::aph* cells exposed to GYY4137 displayed a modest bimodal response as shown by a moderate increase in growth and ATP levels at 10 and 25 μM GYY4137, but significant suppression at higher GYY4137 levels (≥50 μM) (Fig. 4d–f). In contrast, the basal OCR and growth

of GYY4137-treated WT *Mtb* cells remained significantly higher than *cydC::aph* cells at 50 and 100 µM GYY4137 (Figs. 2d, e, 4b, c). In summary, the significantly reduced stimulation of OCR and growth in *cydC::aph* cells (Fig. 4g, h), i.e., when respiration is sustained only by $aa_3$-type cytochrome c oxidase, indicates that H₂S exerts its respiratory and growth-stimulatory effects predominately via CytBD in *Mtb*.

**H₂S modulates sulfur/copper metabolism and the Dos regulon**. To assess the effect of H₂S on the *Mtb* transcriptome, we performed RNA-seq analysis on *Mtb* and *cydC::aph* cells exposed to 25 µM GYY4137 for 1 h. Culture conditions were carefully controlled to eliminate the possibility of respiration-induced hypoxia (see Materials and methods). At 25 µM GYY4137, we observed a distinct transcriptomic profile of 92 differentially regulated genes in *Mtb* compared to untreated cells including those belonging to metabolism and respiration (26/92), and cell wall and cell processes (19/92) (Supplementary Fig. 10A, B). Notably, several members of the CsoR[44] and RicR[45] copper regulons, and six genes involved in cysteine/sulfur metabolism were upregulated (Fig. 5a, b). RNAseq analysis showed no differentially regulated genes in *cydC::aph* cells exposed to 25 µM GYY4137 compared to untreated *cydC::aph* cells, consistent with our observations that H2S exerts its effect primarily through CytBD.

An important finding was that 31 of the 48 genes comprising the *Mtb* Dos dormancy regulon[1], which includes the heme sensor kinase DosS, the response regulator DosR, as well as *hspX, fdxA, narX, narK2, tgs1* and others were significantly upregulated (Fig. 5c). Interestingly, the lack of CsoR was shown to upregulate the Dos dormancy regulon[44]. CsoR is a negative regulator of the copper sensitive operon *cso* whose repression is relieved upon copper binding that leads to the export of copper[46]. We propose that binding of copper to sulfide relieves repression of CsoR, which leads to the export of copper via the *cso* (Fig. 5d).

To determine whether H₂S regulates the Dos regulon via the Dos sensor kinases, we performed qRT-PCR analysis of WT *Mtb* and *Mtb* mutants lacking *dosS* (Δ*dosS*), *dosT* (Δ*dosT*) or both (Δ*dosST*) after exposure to 25 µM GYY4137 for 1 h. Quantitative RT-PCR analysis of Δ*dosS, ΔdosT* and Δ*dosST* strains showed significantly reduced induction of key dormancy regulon genes such as *hspX, fdxA,* and *rv1813c* (Fig. 5e)[1]. These findings are consistent with previous studies showing that the copper regulator, CsoR regulates the Dos regulon[44], and that cysteine stimulates *Mtb* respiration leading to the induction of the CsoR copper and Dos regulons[47].

In sum, our findings provide further credence to the proposed link between sulfide/cysteine-mediated stimulation of respiration, the Dos regulon, and copper homeostasis. The lack of differential gene expression in *cydC::aph* cells compared to cells treated with GYY4137 points to CytBD as an important component of the H₂S energy-transduction pathway. Our data suggest that upon exposure to sulfide, *Mtb* reprograms its metabolism and transcriptional machinery including the copper regulon to mitigate the formation of copper sulfide species. Finally, our data suggest that in addition to •NO, CO and low O₂, we have identified a fourth gas, H₂S, that can induce the Dos dormancy regulon.

**Role of H₂S during *Mtb* growth under hypoxia**. To determine whether growth-stimulatory levels of H₂S accelerate the development of hypoxia-induced, nonreplicating persistence in vitro, we used the anti-TB drug metronidazole (MN), which selectively kills *Mtb* cells under conditions of low O₂[48]. We posited that if H₂S facilitates early entry into hypoxic dormancy, then GYY4137-exposed *Mtb* cells in the presence of MN would be killed more quickly than unexposed cells in the presence of MN. Indeed, we observed a ~20% and ~40% reduction in CFU of

GYY4137-exposed *Mtb* cultures in the presence of MN at 24 and 72 h, respectively (Fig. 5f). In contrast, the viability of unexposed cells was not affected by MN treatment for 24 or 48 h and a slight reduction (~10%) was seen at 72 h. These results show that H₂S can facilitate early entry of *Mtb* cells into hypoxic dormancy. Next, we subjected *Mtb* cultures to hypoxia for 8 weeks with or without initial exposure to GYY4137. Following re-aeration, *Mtb* cells previously exposed to GYY4137 displayed enhanced recovery from hypoxia as shown by more robust growth compared to unexposed cells (Fig. 5g, h). This outcome is consistent with our observation that exposure to low levels of H₂S triggers an antioxidant response evident by upregulation of genes involved in cysteine metabolism (Fig. 5a, b) and antioxidant genes such as *sod*A and *ahp*C[49,50] (Supplementary Fig. 10A), which permit rapid emergence from hypoxia. We subsequently tested the ability of H₂S to protect *Mtb* under conditions of oxidative stress and found that survival of *Mtb* cells exposed to GYY4137 was significantly increased compared to unexposed cells in the presence of the oxidants cumene hydroperoxide (CHP) and menadione (MND) (Fig. 5i). Since H₂S can act as an antioxidant[51,52], H₂S exposure may help mitigate oxidative stress during growth or when *Mtb* emerges from hypoxia.

We next evaluated whether increased respiration following H₂S exposure may result in increased ROS production in *Mtb* cells. We examined the effect of GYY4137 on ROS production during respiration and observed a slight reduction in the percentage of ROS-producing cells following GYY4137 exposure (Fig. 5j), which is likely due to the known antioxidant effect of H₂S. These data are consistent with the growth-promoting effect of H₂S on *Mtb* cells, as excessive ROS production will inhibit growth. We also excluded the possibility that H₂S exerts a direct effect on the ETC via a sulfide quinone oxidoreductase (SQR) since the OCR of the *Mtb sqr* mutant (*rv0331::aph*; TARGET, Johns Hopkins University, USA) exposed to H₂S was similar to that of WT *Mtb* (Supplementary Fig. 11A). This contrasts with *Mtb* exposure to the reductant DTT, which increases the menaquinol pool to stimulate OCR (Supplementary Fig. 11B)[53], but not growth[54]. Overall, our data demonstrate a beneficial role for H₂S during entry and emergence from low-oxygen conditions, as well as protection against oxidative stress.

**Crosstalk between H₂S and •NO alters *Mtb* infection**. Studies in mammalian systems have shown crosstalk between •NO and H₂S[55,56]. We observed increased iNOS levels in the lungs of *Mtb*-infected $Cbs^{+/-}$ mice compared to WT at 8 weeks post infection (Fig. 6a, b). Consistent with increased iNOS levels, infected $Cbs^{+/-}$ mice had higher $NO_2^- + NO_3^-$ levels in serum compared to WT mice, suggesting increased •NO production in $Cbs^{+/-}$ mice (Fig. 6c). Increased $NO_2^- + NO_3^-$ levels were also observed in the culture media of infected $Cbs^{+/-}$ macrophages (Fig. 6d). Addition of GYY4137 significantly lowered $NO_2^- + NO_3^-$ levels in WT and $Cbs^{+/-}$ macrophages (Fig. 6d). These observations are consistent with the ability of H₂S to regulate •NO bioavailability in vivo, including during inflammation[57,58]. By itself, •NO exerts bacteriostatic effects on *Mtb* by inhibiting respiration[1], and increased •NO may also contribute to inhibition of bacterial respiration and growth during infection. As expected, exposure of *Mtb* to •NO in vitro decreased basal OCR (Fig. 6e, f). Intriguingly, the OCR was restored following exposure to GYY4137 (Fig. 6e, f). ATP levels were also reduced in *Mtb* cells following exposure to •NO (Fig. 6g), suggesting inhibition of respiration[1]. In contrast, the ATP levels of cells exposed first to •NO followed by GYY4137 were similar to those in unexposed cells, corroborating our observation that GYY4137 restored OCR following •NO exposure (Fig. 6e, f). We then performed growth

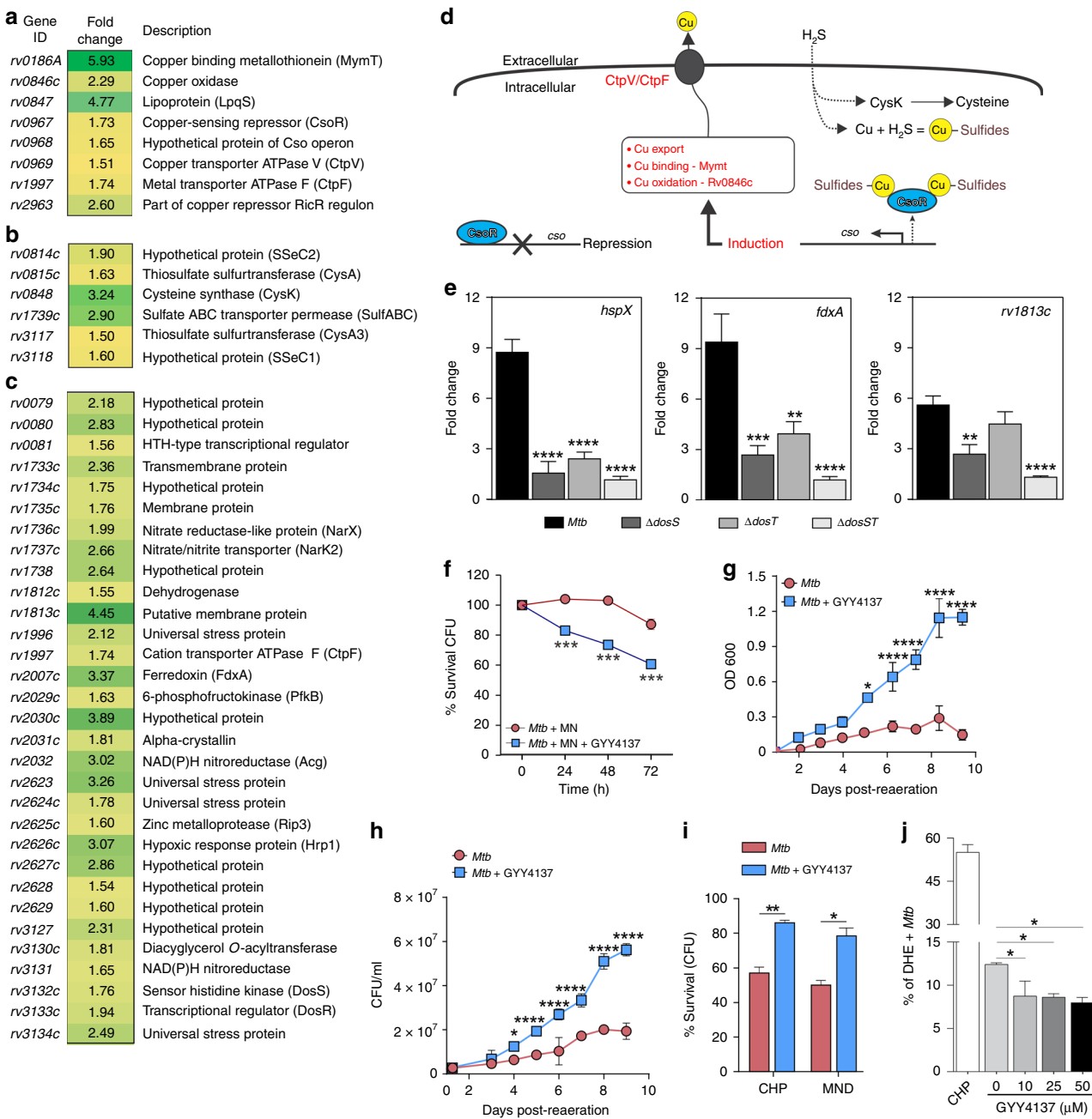

**Fig. 5 H₂S induces transcriptome remodeling and mitigates oxidative stress in *Mtb*.** RNA-seq-based gene expression analysis shows induction of the *Mtb* **a** copper, **b** sulfur and **c** Dos regulons following exposure to 25 μM GYY4137 for 1 h. Genes exhibiting ≥1.5-fold (+/−) change in expression compared to the expression in untreated cells were considered differentially regulated (*n* = 3 independent biological replicates per group). **d** Proposed model detailing the relationship between H₂S and induction of the *Mtb* copper regulon. Exposure to H₂S leads to the formation of copper sulfides, which relieve repression of CsoR leading to the export of copper via the *cso*. **e** Representative results of qRT-PCR analysis of key dormancy regulon genes following 1 h exposure to 25 μM GYY4137 in *Mtb* H37Rv, Δ*dosS*, Δ*dosT* and Δ*dosST* (*n* = 5 independent biological replicates per *Mtb* strain). Fold change in gene expression following addition of GYY4137 is shown. **f** Survival of *Mtb* cells (% CFU) exposed to metronidazole (MN) alone, or MN and 25 μM GYY4137. **g** Recovery profile of *Mtb* cells following prolonged hypoxia (8 weeks). Prior to hypoxia, one group of cells received 25 μM GYY4137. **h** Mean CFUs following 1 week of recovery following prolonged hypoxia. (**g**) and (**h**) represent pooled data from two or more independent experiments (*n* = 4). **i** Percentage survival (CFU) of 25 μM GYY4137-treated *Mtb* cells following exposure to oxidative stressors CHP and MND (0.5 mM for 6 h) relative to untreated cells. **j** Percent of dihydroethidium (DHE)-positive ROS-producing *Mtb* cells following exposure to 10−50 μM GYY4137 (spent GYY4137 used as 0 μM control; *n* = 3 per group). CHP treatment (15 mM) was used as a positive control for detection of ROS-positive cells. Data in (**f**) and (**i**) are representative of two or more experiments. All data are shown as mean ± SD, except (**e**) (mean ± SEM). Data were analyzed using one-way ANOVA (**e** and **j**) with Sidak's multiple comparisons test, or by two-way ANOVA with Sidak's multiple comparisons test (**f**−**h**), or by unpaired parametric *t* test (**i**). *$P < 0.05$, **$P < 0.01$, ***$P < 0.001$, ****$P < 0.0001$. Source data are provided as a Source Data file.

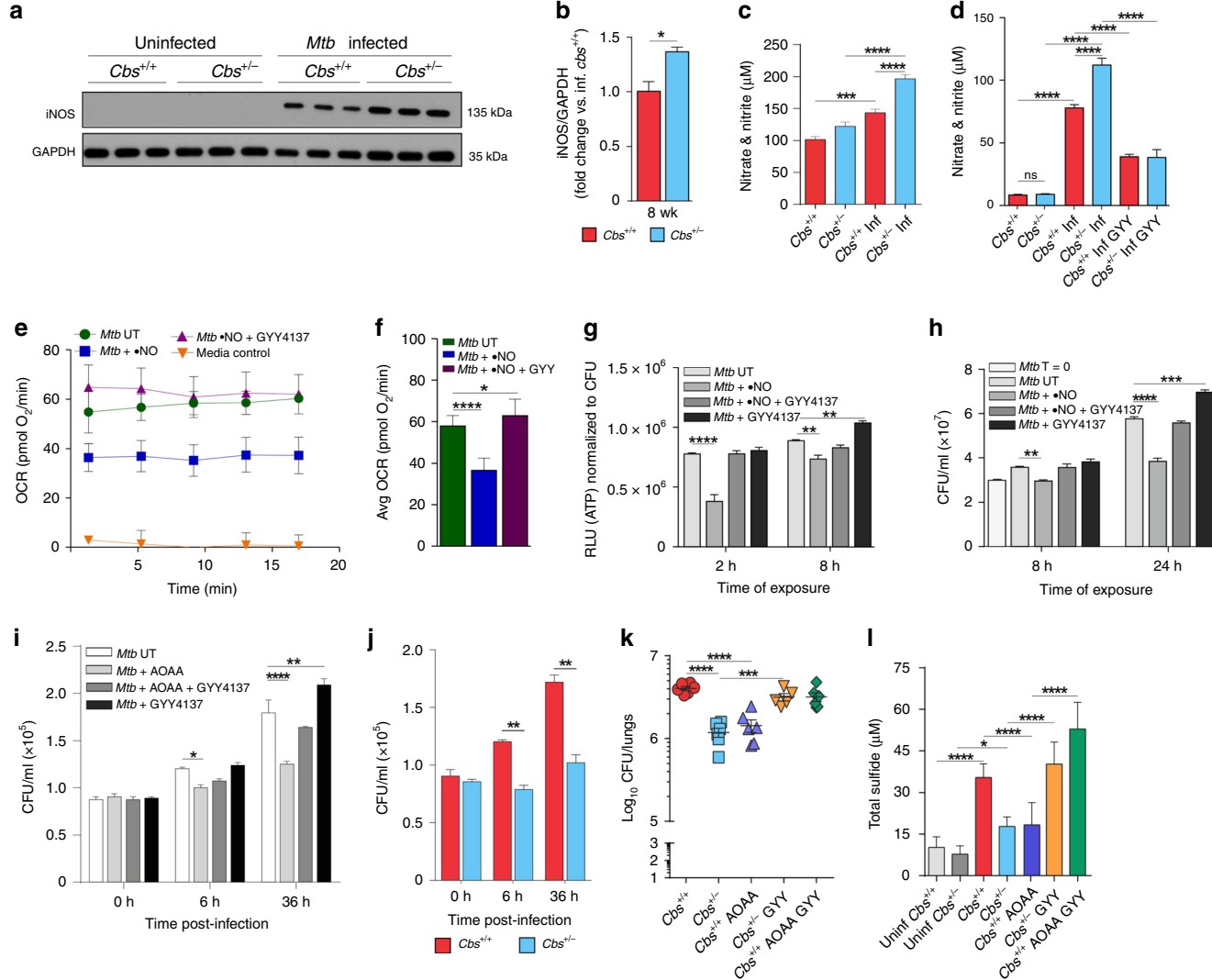

**Fig. 6 Interaction between •NO and H₂S and pharmacological modulation of sulfide levels in mice. a** Representative western blot and **b** densitometry analysis of iNOS in mouse lungs at 8 weeks post infection ($n = 3$ animals per group). **c** Nitrate and nitrite levels in the sera of infected (Inf) and uninfected mice and **d** the culture medium of infected mouse macrophages exposed to 25 μM GYY4137 for 36 h. Data in (**c**) and (**d**) are pooled from two independent experiments ($n = 7-8$ animals per group). **e** OCR profile and **f** average OCR of *Mtb* cells exposed to •NO (DETA/NO; 250 μM), •NO followed by GYY4137 (10 μM), or untreated (UT) in XF media ($n = 6-8$). Media control contained no cells. **g** ATP levels (shown as RLU normalized to CFU) in *Mtb* cells treated (•NO, GYY4137, or •NO and GYY4137) or untreated (UT) for 2 or 8 h. **h** CFUs after 8 and 24 h of exposure. $T = 0$ indicates CFU at the start of the experiment. Data in (**g**) and (**h**) are representative of at least two independent experiments. **i** CFU recovered from infected *Cbs⁺/⁺* peritoneal macrophages receiving one of the following: aminooxyacetic acid (AOAA), AOAA followed by GYY4137, GYY4137 alone, or untreated (UT) ($n = 4$). **j** *Mtb* CFU recovered from mouse peritoneal macrophages. **k** Bacillary burden in the lungs of mice that received: AOAA, AOAA followed by GYY4137, or GYY4137 alone. AOAA (13 mg/kg, 6 d/wk for 6 weeks) and GYY4137 (25 mg/kg 3 d/wk for 6 weeks) were administered intraperitoneally. Treatment began 2 weeks post-*Mtb* infection and continued for 6 wks. Data represent two independent experiments ($n = 5-6$ animals per group). **l** Total sulfide levels in serum of mice in (**k**) using the methylene blue method ($n = 6-8$). Data represent mean ± SEM except in (**e**, **f** and **l**) (mean ± SD). Comparisons are against WT mice or untreated (UT) controls. Data were analyzed by unpaired parametric *t* test (**b**, **j**), one-way ANOVA with Tukey's multiple comparisons test (**c**, **d**, **f**, **k** and **l**) or two-way ANOVA with Sidak's multiple comparisons test (**g**–**i**). *$P < 0.05$, **$P < 0.01$, ***$P < 0.001$, and ****$P < 0.0001$. Source data are provided as a Source Data file.

assays using *Mtb* cells exposed to •NO alone, GYY4137 alone, or •NO and GYY4137. *Mtb* cells exposed to GYY4137 exhibited significantly increased growth whereas growth was reduced in cells exposed to •NO, confirming the bacteriostatic effect of •NO (Fig. 6h). When GYY4137 was added following •NO exposure, the CFU recovery was similar to that of unexposed cells (Fig. 6h). While the exact chemical nature of the crosstalk between H₂S and •NO remains poorly defined, these data suggest that exposure to H₂S reverses the bacteriostatic effect of •NO on *Mtb* respiration and growth.

**Pharmacological inhibition of CBS in vitro and in vivo.** To validate our in vitro findings that H₂S stimulates *Mtb* respiration and growth, we tested whether manipulating H₂S levels via pharmacological modulation of H₂S-producing enzymes could alter disease progression. First, we infected peritoneal macrophages from WT mice and treated the cells with the CBS inhibitor aminooxyacetic acid (AOAA)[14]. We observed ~17% and ~30% reductions in CFU from infected macrophages after exposure to AOAA for 6 and 36 h, respectively (Fig. 6i). In contrast, CFU recovery from AOAA-treated WT macrophages exposed to

GYY4137 was comparable to recovery from untreated macrophages indicating that $H_2S$ itself is primarily responsible for the reduced bacterial burden in AOAA-treated cells (Fig. 6i). Further, within *Mtb*-infected $Cbs^{+/-}$ peritoneal macrophages, bacillary growth was reduced by ~35% and 40% versus WT cells at 6 and 36 h post infection, respectively (Fig. 6j).

Next, we show that administration of the CBS inhibitor AOAA to WT mice reduces *Mtb* burden in the lungs to the level of $Cbs^{+/-}$ mice (Fig. 6k). Further, administration of GYY4137 to $Cbs^{+/-}$ mice increased *Mtb* burden to that of WT mice. Of note, the bacillary burden in the lungs of WT mice treated with AOAA and GYY4137 was similar to the burden in untreated WT mice (Fig. 6k). Determination of sulfide levels in the sera of these same groups of mice revealed that *Mtb* infection increased sulfide levels (Fig. 6l). Further, we observed that the reduction in serum sulfide observed in $Cbs^{+/-}$ mice and AOAA-treated WT mice was associated with reduced lung bacillary burden (Fig. 6k, l). CFU recovery in $Cbs^{+/-}$ mice and AOAA-treated WT mice following GYY4137 treatment was comparable to that in untreated WT mice, confirming our previous findings that differences in CFU are due to $H_2S$, and not CBS. While it could be argued that AOAA targets *Mtb* directly, addition of AOAA to *Mtb* cultures at concentrations as high as 1 mM does not inhibit *Mtb* growth. However, 2 mM AOAA reduces *Mtb* proliferation by ~35% (Supplementary Fig. 12). Thus, while our in vitro and in vivo infection data demonstrate that AOAA reduces host $H_2S$ production to modulate *Mtb* infection, we cannot completely exclude the possibility of a direct effect of AOAA on *Mtb*. Overall, our findings provide evidence that $H_2S$ is a modulator of *Mtb* respiration, and that host-derived $H_2S$ is a significant contributor to TB disease.

## Discussion

Here we establish a previously unrecognized role for $H_2S$ in promoting TB, wherein increased host $H_2S$ levels following *Mtb* infection promote bacterial growth, thereby reducing host survival. In addition, we have identified a genetic locus, *Cbs*, which exacerbates TB disease. This was evident from our mouse studies that demonstrated significantly increased survival of infected $Cbs^{+/-}$ mice, which produce reduced levels of $H_2S$. This is an unusual finding as studies to date have shown that disruption of host loci implicated in TB lead to increased susceptibility and not resistance. Our findings suggest that *Mtb* elicits a deleterious host response via CBS to promote progressive pulmonary disease. Our data show that $H_2S$ stimulates *Mtb* respiration, predominately via CytBD, and reprograms central metabolism leading to enhanced growth. We observed that iNOS expression is higher in infected $Cbs^{+/-}$ mice and that $H_2S$ effectively reverses •NO-mediated suppression of *Mtb* respiration in vitro. Further, pharmacological inhibition of CBS by AOAA reduced the sulfide levels and bacillary load in mice, and this effect could be reversed by administration of GYY4137, a physiologically relevant $H_2S$ donor. Since host $H_2S$ biosynthesis does not require $O_2$ as a cofactor[59,60] and $H_2S$ could function as an *Mtb* fuel source during hypoxia, our findings have implications for understanding how *Mtb* can survive for extended periods under hypoxic conditions. Overall, our findings suggest a feedback loop mechanism, wherein host $H_2S$ production is triggered by *Mtb* to promote a hypermetabolic state that supports establishment of TB disease.

The role of host-generated $H_2S$ in microbial pathogenesis is poorly understood. To the best of our knowledge, no study has yet reported a role for host $H_2S$ in the control of bacterial disease. A biphasic response to $H_2S$ is well established wherein $H_2S$, depending on concentration, can trigger a variety of physiological and immunological responses, which can be cytotoxic or cytoprotective[14,16,61]. We initially found that CBS protein and sulfide levels were increased in *Mtb*-infected macrophages, suggesting a direct link between *Mtb* infection and $H_2S$. Further studies showed that $Cbs^{+/-}$ mice exhibited lower sulfide levels that corresponded with reduced bacterial burden and increased survival. We then sought to determine whether differential immune responses accounted for the increased survival and reduced bacillary burden in $Cbs^{+/-}$ mice. Given the relative minor differences in immune response between WT and $Cbs^{+/-}$ mice, and that $H_2S$ can act directly on bacteria, we posited that differences in bacillary burden and survival are due to the direct action of host $H_2S$ on *Mtb* physiology. Indeed, we observed modest differences in organ burden between $Cbs^{+/-}$ and WT mice, but a striking difference in survival (Fig. 1g). Such a disparity between organ burden and survival is not unusual and was previously reported for *Mtb* whiB3[62] and sigH[63] mutants that showed similar organ burden, but substantially prolonged survival. Overall, our findings suggest that *Mtb* possesses yet-to-be identified mechanisms that trigger upregulation of CBS, ultimately leading to increased organ burden and mortality.

Similar to the effects of $H_2S$ observed in mammalian cells[14], we show that low levels of $H_2S$ exert a dose-dependent bimodal effect on *Mtb* growth and promote growth by stimulating respiration and ATP production. We confirmed the ability of $H_2S$ to stimulate respiration and growth of *Mtb*, *M. bovis* BCG and *M. smegmatis* using two independent respirometry techniques and two different $H_2S$ donors. Our results are complementary to recent studies of *E. coli*, a facultative anaerobe and member of the gut microbiota that is continuously exposed to large quantities of $H_2S$[38,64]. These studies showed that $H_2S$ inhibits the *E. coli* heme-copper $bo_3$ oxidase whereas the two cytochrome BD oxidases are resistant to $H_2S$. An unusual feature of *Mtb* respiration and metabolism is the ability to be stimulated by $H_2S$. Using respirometry, RNAseq and growth assays, our data indicate that cytochrome $bc_1$-$aa3$ (fused complex III/IV) is the primary target of $H_2S$ and that CytBD mediates the outcome of metabolic stimulation and growth.

Initial studies of the stimulatory effect of $H_2S$ on mitochondrial respiration have shown that at low sulfide levels, $H_2S$ acts as a substrate by binding and reducing cytochrome c oxidase (Cco)[65,66]. The cytochrome c oxidase complex contains two redox active metal sites, heme *a* and $Cu_A$, that donate electrons to the binuclear center composed of heme *a3* and $Cu_B$, which reduces $O_2$[67,68]. The redox states of the heme prosthetic groups and copper centers play a critical role in their reactions with gases and can mediate crosstalk between gaseous ligands. For example, in contrast to •NO and CO, which bind primarily to the binuclear center in the fully reduced ($Fe^{2+}/Cu^{1+}$) state, sulfide does not react with the fully reduced enzyme, but reacts with met ($Fe^{3+}$) heme. Hence, reactions with sulfide occur in the oxidized or mixed valence states[55,69]. A recent model for sulfide-mediated inhibition of cytochrome c oxidase proposed that both $Fe^{3+}$ heme *a3* and $Cu_B$ ($Cu^{1+}$ or $Cu^{2+}$) can bind $H_2S$ or $SH^-$[69]. The final enzymatic product of sulfide inhibition was suggested to be a mixed valence form of the enzyme containing reduced heme *a* and $Cu_A$, whereas heme *a3* is in a ferric sulfide-bound state, which blocks the reaction with $O_2$[66,70]. A second sulfide molecule is also possibly bound to $Cu_B^{+}$[15,69]. It should be noted that both $CN^-$ and $H_2S$ bind heme iron in the $Fe^{3+}$ state and not surprisingly, also compete for the heme $a_3$-$Cu_B$ binuclear center[71,72]. Regardless of the precise mechanism whereby $H_2S$ regulates *Mtb* respiration and metabolism, our data suggest that *Mtb* cytochrome $bc1/aa3$ is a target for host-generated $H_2S$. Lastly, the mechanisms whereby $H_2S$ stimulates *Mtb* respiration and growth are expected to be different as *Mtb* has the remarkable capacity to rapidly reroute electrons to cytochrome $bc_1$-$aa3$ or CytBD[30], and

the ETC is not subjected to backpressure. While it is possible that H2S stimulates *Mtb* respiration via an SQR, our data suggest this is unlikely since lack of *rv0331* did not alter the OCR following H2S exposure. However, more studies are needed to conclusively demonstrate whether Rv0331 is indeed an SQR.

Since respiration is tightly linked to metabolism, we expected that H2S-induced changes in *Mtb* respiration would impact central metabolism. Indeed, our carbon tracing and FNC analyses show that H2S stimulates the PPP, which provides nucleosides for DNA synthesis and NADPH for lipid anabolism, which is required for increased metabolism and growth and the antioxidant response. Of interest is the capacity of H2S to rewire energy flux by increasing NADH and ATP levels, and to control the carbon fractions of glycolytic and TCA metabolites, pointing to the complexity of this gas's interaction with central metabolism.

The stimulation of respiration and upregulation of the Dos regulon by H2S highlight an unusual facet of *Mtb* physiology. For example, inhibition of *Mtb* Complex III/IV (cytochrome *bc1-aa3* oxidase) and Complex V (ATP synthase) with the anti-TB drugs Q203 and bedaquiline (BDQ), respectively, dramatically enhances *Mtb* respiration[30]. Furthermore, increased respiration[30] and induction of the Dos regulon by BDQ[73] or cysteine[47] provide evidence of a link between respiratory control and the Dos regulon. Paradoxically, NO, CO, and low O2 inhibit respiration and bind to DosS/DosT heme irons to induce the Dos regulon[1,4,5], whereas H2S, BDQ and cysteine stimulate respiration leading to induction of the Dos regulon. These findings raise the questions of how H2S induces the Dos regulon and what the relationship is between sulfur and copper metabolism as identified in our RNAseq experiments. Since H2S can coordinate with heme[65], it is possible that H2S directly triggers upregulation of the Dos regulon via the heme-containing sensor kinases DosS and/or DosT. However, the DosT heme iron is in the oxy (O2) state and unable to bind H2S whereas DosS is in the $Fe^{3+}$ state[74], which can bind H2S. The precise mechanism of regulation of the Dos regulon appears to be more complex than previously anticipated and is confounded in part by the lack of knowledge of how in vitro biochemical data relate to the redox states of DosS and DosT in the *Mtb* cell.

By examining our expression data, it could be argued that induction of the Dos dormancy regulon by H2S is less robust than that of •NO[1] or hypoxia[5]. However, a direct comparison is not possible since different concentrations of gases were used and their diverse physicochemical properties will influence binding to DosS/DosT heme irons (i.e., solubility, diffusion, on- and off-rates, etc.). Notwithstanding, the overall levels of induction of Dos dormancy regulon members by H2S is comparable to[4] and may exceed that of CO[75].

Upregulation of the Dos regulon in a *Mtb* csoR mutant has been reported[44]. Based on the unusually strong affinity between H2S and copper, we posit that copper is (i) exported via the copper responsive proteins CtpV[76] and CtpF, (ii) bound via the strong copper-binding protein metallothionein [MymT;[77]], and (iii) oxidized by the copper oxidase (*rv0846c*). Further attempts by the cell to detoxify excess H2S are evident by the upregulation of *rv0848* that detoxifies sulfide to generate cysteine (*O*-acetyl-L-serine + H2S → L-cysteine + acetate). We propose that accumulation of toxic copper sulfides relieve CsoR-mediated repression of the *cso* regulon to promote copper export.

It is difficult to ignore the critical role that the copper-containing cytochrome *bc1/aa3* plays in the induction of the Dos dormancy regulon as it is targeted by •NO, CO, and O2 (hypoxia) to inhibit respiration and induce the Dos regulon. H2S generates a distinct response since it reroutes electrons away from cytochrome *bc1/aa3* toward the non-copper-containing CytBD, but also induces the Dos dormancy regulon. Intriguingly, this model

may not appear to agree with a study showing induction of the Dos dormancy regulon with the anti-TB drug BDQ[73], which targets *Mtb* Complex V and not cytochrome *bc1/aa3*. However, this drug also reroutes electrons to CytBD[30]. Hence, it appears that the rerouting of electrons to CytBD, and therefore the *Mtb* ETC, plays a more prominent role in the induction of the Dos regulon that previously appreciated. One limitation of this study is that currently available pharmacological inhibitors of CBS (such as AOAA) have suboptimal bioavailability and a degree of nonspecificity[78], which limit potency and clinical application. Further, as is evident from the wide range of H2S levels reported in the blood of mice (0.1−80 μM) and humans (2−110 μM)[19], accurate measurement of H2S concentrations in biological samples is problematic due to the complex sulfur chemistry and extreme volatility of H2S. In fact, critical assessment of the literature and H2S chemistry suggests that H2S concentrations in vivo are in the nanomolar range[19–21]. Not surprisingly, there is a lack of a universally accepted gold standard for the measurement of low concentrations of H2S[19–21]. A constraint of the commonly used methylene blue-based assay for H2S quantitation is that it measures total sulfide, not H2S alone, and can lead to overestimation of H2S concentrations. In contrast, the H2S-specific fluorogenic probe-based approach circumvents some of the limitations of the methylene blue method, but provides only relative quantitation[20]. Clearly, methods to measure H2S accurately and consistently in human tissue, fluids or biological cultures are required to further explore a role for H2S in TB. Lastly, generating tissue- or cell-type-specific *Cbs* knockout mice should allow more precise functional studies of the role of CBS in TB.

We demonstrate that exploiting host H2S is an important *Mtb* survival strategy to establish disease (Fig. 7). These findings add to our understanding of the complex interaction between host gasotransmitters and *Mtb* and suggest the potential of pharmacological targeting of H2S production as a therapeutic strategy for TB treatment. The interplay between H2S and •NO that likely modulates TB pathophysiology is reflected in our animal studies and points to complex interactions at the primary site of infection, the lung. Lastly, knowing the spatial distribution of H2S-producing enzymes in human hypoxic TB lesions may have important implications as H2S possibly modulates *Mtb* bioenergetics under hypoxic conditions. It is tempting to speculate that H2S may function as an essential fuel for maintaining *Mtb* energy metabolism during long-term persistence in hypoxic human TB lesions.

## Methods

**Study design**. In this study, we hypothesized that host-derived H2S modulates *Mtb* pathogenesis via respiratory and bioenergetics changes. To test this hypothesis, we first used *Cbs*-deficient mice (*Cbs*[+/−]) to study survival and *Mtb* disease progression, and determined immunological changes. To determine the effect of H2S on *Mtb* physiology, we studied *Mtb* growth kinetics, gene expression, metabolism and respiration. All animal experiments were conducted according to the highest ethical standards and approved by the Institutional Animal Care and Use Committee (IACUC) of the University of Alabama at Birmingham.

**H2S donor compounds and preparation**. GYY4137 [morpholin-4-ium 4 methoxyphenyl (morpholino) phosphinodithioate] (Cayman Chemicals, USA) and NaHS (Alfa Aesar, USA) were used as sources of H2S and were prepared according to the manufacturer's instructions. While NaHS releases H2S instantaneously, GYY4137 is a water soluble H2S donor that releases H2S slowly for weeks both in vitro and in vivo[24,27], and more closely parallels the biological effect of endogenous H2S. For H2S exposure experiments, a 10 mM GYY4137 stock solution was prepared in water in a 5 ml tube and was immediately diluted to make a working stock and added to cultures to achieve the desired final concentration. Experimental control samples received time-spent 25 μM GYY4137 (referred to as spent GYY) that was previously aerated for at least 120 days. An NaHS stock solution (100 mM) was prepared by equilibrating phosphate-buffered saline (PBS), made anoxic by bubbling with argon, with NaHS in a septum-sealed glass syringe. In order to avoid any trace metals catalyzing the oxidation of H2S, the metal chelator DTPA (diethylenetriaminepentaacetic acid) was added to the buffer to a final

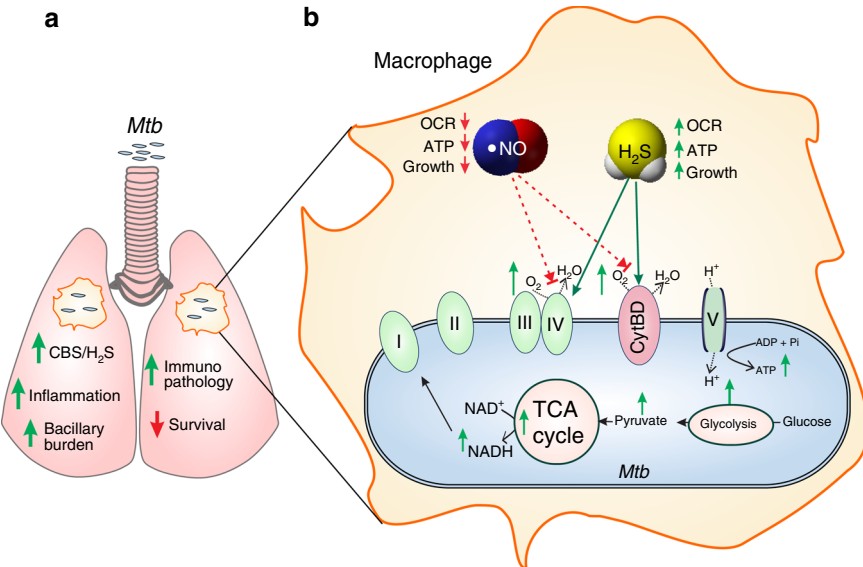

**Fig. 7 Proposed mechanism for H₂S-induced disease exacerbation. a** As a response to *Mtb* infection, the host increases CBS and H₂S production in the lungs and macrophages leading to increased bacillary burden, increased inflammation, and reduced survival. **b** Host-generated H₂S is sensed by *Mtb* to stimulate respiration and energy metabolism, thereby increasing bacterial growth and accelerating disease. During infection, increased H₂S suppresses iNOS levels and production of •NO. H₂S also mitigates the bacteriostatic effect of •NO on *Mtb* respiration. Green arrows represent increase; red arrows represent decrease.

concentration of 50 μM prior to addition of NaHS. Photo-oxidation was prevented by wrapping the solution-containing syringe with aluminum foil. Despite these precautions, the presence of polysulfide in stock solutions cannot be completely excluded.

**Bacterial strains and growth conditions.** *M. tuberculosis* H37Rv (obtained from BEI Resources (NR-123), *Mtb* cydC::aph (containing a transposon in *cydC*; kindly provided by Dr. Digby Warner, University of Cape Town), *Mtb* ΔdosS, ΔdosT and ΔdosS/dosT mutants (kindly provided by Dr. David Sherman, University of Washington), *M. smegmatis* mc²155 and *M. bovis* BCG were grown in Difco Middlebrook (MB), 7H9 (broth) or BBL7H11 agar (BD Biosciences, USA) supplemented with 0.2% glycerol, 1× ADS (albumin, dextrose, sodium chloride) and 0.02% tyloxapol at 37 °C. Complementation of *Mtb* cydC::aph was achieved by placing *cydC* under control of the *hsp₆₀* promoter. Cloning the *hsp₆₀-cydC* fragment into a stably integrating *E. coli*-mycobacterial plasmid (hygromycin resistance) followed by transformation and selection yielded viable *Mtb* cydC::aph-Comp colonies that showed a mild growth delay. H₂S donor compounds were prepared as mentioned above and added to the required concentration for growth assays. Cultures were removed from sealed vials at specific time points, serially diluted and aliquots were plated on BBL 7H11 agar plates. All sealed culture vessels containing H₂S donors were only opened once. CFU were enumerated after 4−6 weeks of incubation at 37 °C.

**Breeding and genotyping of mice.** Original colonies of C57BL/6J (*Cbs⁺/⁺*) and B6;129P2-Cbs^tm1Unc/J heterozygous mice (*Cbs⁺/⁻*) were obtained from the Jackson Laboratory (stock# 000664 and 002461, respectively; Bar Harbor, ME) and subsequently backcrossed for at least seven generations before starting the experiment[22]. Genotypes of *Cbs⁺/⁺* and *Cbs⁺/⁻* mice were confirmed by performing standard PCR from digests of tail snips. The following diagnostic primers were used for genotyping according to Jackson Laboratory guidelines: a common Fwd WT allele primer 5′-GATTGCTTGCCTCCCTACTG-3′ (Jackson primer ID 12998); WT allele Rev primer 5′-AGCCAACTTAGCCCTTACCC-3′ (Jackson primer ID 12999) and mutant allele Rev primer 5′-CGTGCAATCCATCTTGTTCA-3′ (Jackson primer ID 12400). Food (standard chow) and water were provided ad libitum.

***Mtb* infection of mice.** Male and female age-matched mice aged 8−10 weeks were infected with *Mtb* H37Rv either by intratracheal instillation or by an aerosol inhalation exposure system (Glas-Col, USA). Lung CFU at 24 h post infection was used to determine the initial dose of infection. For time course infection studies, mice were sacrificed at 3, 8 and 14 weeks post infection. The number of viable bacteria present in lungs and spleens was enumerated by plating serial dilutions of homogenates on BBL 7H11 agar plates supplemented with carbencillin (50 μg/ml) and cycloheximide (50 μg/ml). CFU were enumerated as described above. For survival studies, 10 mice/group were used, and time-to-death was observed and represented by a Kaplan−Meier graph. Mice were monitored for survival until

300 days whereas all *Cbs⁺/⁻* mice died by day 270. No deaths were observed in uninfected WT or *Cbs⁺/⁻* mice within 1 year of the end of the survival experiment.

**AOAA/GYY4137 treatment in mice.** Mice were infected with *Mtb* H37Rv as described earlier by an aerosol inhalation exposure system (Glas-Col, USA). The number of lung CFU at 24 h post infection was used to determine initial dose of infection. After 2 weeks of initial infection, mice were subjected to intraperitoneal administration of AOAA (13 mg/kg/day, 6 d/wk for 6 weeks), GYY4137 (25 mg/kg/day, 3 d/wk for 6 weeks) or both. Control mice received vehicle control (PBS). Treatment started 2 weeks post-*Mtb* infection and lasted for 6 weeks when mice were sacrificed (*n* = 6). The number of lung CFU was determined as described above.

**Histological analysis of mouse lung lesions.** Lung lobes were fixed in 10% neutral buffered formalin for a minimum of 48 h then embedded in paraffin by standard procedures. Tissues were processed and stained at the UAB Comparative Pathology Laboratory. Briefly, paraffin sections were cut at a thickness of 5 μm with a rotary microtome. Sections were stained with hemotoxylin and eosin to score for pathology. Microscopic images were taken and NIS Elements software (Nikon) was used for semiquantitative morphometric analysis. For each slide, multiple representative, but random microscopic fields were analyzed. An experienced pathologist calculated the percentage surface area covered by lesions between *Cbs⁺/⁺* and *Cbs⁻/⁻* mice. Micrographs were taken at ×10 magnification.

**Cytokine analysis.** Mouse blood was obtained using protocols approved by the UAB IACUC and sera isolated by standard methods. Levels of Th-1 and Th-2 cytokines were determined using a Bio-Plex Pro mouse cytokine 17-plex kit (Bio-Rad, USA), following the manufacturer's recommendations.

**Isolation of macrophages.** Peritoneal macrophages were isolated from mice using aseptic technique. Briefly, mice were injected with 1 ml of 3% Brewer thioglycollate medium into the peritoneal cavity. Three days after injection, elicited macrophages were collected from the peritoneal cavity by injecting and withdrawing 5 ml of sterile PBS. Red blood cells were removed by lysis with ACK lysing buffer followed by washing with PBS pH 7.4.

**Macrophage infections and AOAA/GYY4137 treatment.** *Mtb* cells were cultured to mid-log phase (OD₆₀₀ ~ 0.8) in MB 7H9 medium, pelleted by centrifugation, washed once in PBS Tween (PBST) and resuspended in PBST. *Mtb* cells were passed through 27-gauge needle ten times to insure a uniform suspension for infection. Mouse peritoneal macrophages (1.5 × 10⁵) were infected at a multiplicity of infection (MOI) of 1:1 (bacilli:macrophage). Following infection, extracellular bacteria were removed by treatment with amikacin (200 μg/ml) for 2 h followed by washing. The macrophages at this stage received: AOAA alone (2 mM), GYY4137 alone (25 μM) or AOAA + GYY4137. To determine the intracellular growth of *Mtb*, macrophages were harvested and lysed by the addition of 0.025% SDS (Sigma,

USA) at specific time points post infection. Appropriate dilutions of the lysates then were plated on MB7H11 agar. Colonies were counted after 4 weeks of incubation at 37 °C and data expressed as CFU/ml.

**Immunoblotting.** RAW264.7 macrophages (ATCC, USA) or murine peritoneal macrophages were cultured in complete RPMI media and were infected at an MOI of 1:1 (bacilli:macrophage) as described[29]. Uninfected cells were used as controls. Following 6 h of *Mtb* infection, RAW cell monolayers were washed twice with PBS and whole cell lysate was prepared in RIPA lysis buffer (50 mM Tris/HCl, 1% NP-40, 0.25% deoxycholic acid, 150 mM NaCl, 1 mM EGTA, 1 mM sodium orthovanadate, and 1 mM sodium fluoride) containing a protease inhibitor cocktail tablet (Roche, USA) as per the manufacturer's instructions. Total protein was quantified using bicinchoninic acid protein assay (Thermo Fisher Scientific, USA). Twenty micrograms of total protein was used for analysis and was resolved on 4−20% gradient Tris-glycine SDS-polyacrylamide gels (Bio-Rad, USA) and transferred to a poly-vinylidene fluoride membrane (Bio-Rad). Membranes were blocked with 5% non-fat dry milk in Tris-buffered-saline (TBS) with Tween (0.1%) for 1 h and then incubated with primary antibodies directed against CBS (1:1000; Santa Cruz; sc-67154), 3-MST (1:500; Abcam; ab85377), CSE (1:2500; Proteintech; 12217-1-AP) or GAPDH (1:5000; Santa Cruz; sc-25778) followed by a peroxidase-conjugated goat anti-rabbit IgG secondary antibody (1:20,000; Abcam; ab6721). Horseradish peroxidase activity was detected using an enhanced chemiluminescence detection system (GE Healthcare, USA).

Lungs isolated from *Mtb*-infected and uninfected age-matched mice were independently processed and were homogenized in RIPA buffer. Twenty micrograms of total protein was resolved as described above. Membranes were blocked as described above and then incubated with primary antibodies directed against CBS (1:1000; Santa Cruz; sc-67154), iNOS (1:1000; Abcam; ab3523), or GAPDH (1:5000; Santa Cruz; sc-25778), followed by a peroxidase-conjugated goat anti-rabbit IgG secondary antibody (1:20,000; Abcam; ab6721). Horseradish peroxidase activity was detected as described above. Densitometry was performed using ImageJ software and results were normalized against GAPDH expression. Uncropped, unprocessed scans of immunoblots are included in the Source Data file.

**Flow cytometry.** Single-cell suspensions were prepared from lungs and spleens by finely dicing the tissue followed by digestion with liberase DL (Roche, USA) at 37 °C for 30 min. RBC were removed by treatment with ACK lysis buffer followed by washing with PBS pH 7.4. Isolated cells were divided in two groups for myeloid and lymphoid cell characterization. For surface staining, cells were treated with fixable viability dye (eBiosciences, USA) for 30 min at 4 °C in the dark. Fc receptors were blocked with anti-CD16/32 for 10 min at 4 °C in the dark. Immune cell populations were identified by immunostaining using fluorescence-conjugated antibodies directed against the cell-surface markers for 30 min at 4 °C. For intracellular staining, prior to surface stain, cells were cultured in RPMI-1640 containing 5 ng/ml PMA, 500 ng/ml Ionomycin (Sigma, USA) and 10 μg/ml GolgiPlug protein transport inhibitor (BD Biosciences, USA) at 37 °C for 4 h. Cells were then harvested, washed twice for cell-surface markers followed by permeabilization using a BD cytofix/cytoperm permeabilization kit (BD Biosciences) and then stained for intracellular IFN-γ. The following fluorescence-conjugated antibodies were used: Ly-6G (Gr-1) APC (clone: 1A8-Ly6g; eBioscience # 17-9668-82; 10 μl/200 μl reaction), CD11b eVolve™ 605 (clone: M1/70; eBioscience # 83-0112-42; 5 μl/200 μl reaction), F4/80 Antigen APC-eFluor® 780 (clone: BM8; eBioscience #47-4801-82; 10 μl/200 μl reaction), CD4 eVolve™ 605 (clone: RM4-5; eBioscience # 83-0042-42); CD45.2-Bv650 (clone: 104; BioLegend #109836; 10 μl/200 μl reaction); IFN-γ-APC (clone: XMG1.2; BioLegend # 505810; 10 μl/200 μl reaction). Finally, cells were washed twice followed by fixation using IC-fixation buffer (eBiosciences) before analysis. Flow cytometry acquisitions and analyses were performed using Becton Dickinson LSR II with FACS Diva 8.0 software (BD Biosciences, USA). Data were further analyzed using FlowJo 10 (Tree Star, USA).

**Bioenergetic analysis using the Seahorse XF96 analyzer.** OCR profiles of mycobacterial strains adhered to Cell-Tak®-coated XF cell culture microplates at $2 \times 10^6$ bacilli/well were measured using a XF96 Extracellular Flux Analyzer (Agilent Technologies, USA) as described previously[29–31]. The basal oxygen consumption (OCR) rate of bacterial cells was measured with or without GYY4137 and the shift in the OCR profile was evaluated. In assays using spermine NONOate (Cayman Chemicals, USA) and GYY4137, cells were exposed overnight (or injected during the XF96 run depending on the experiment) to spermine NONOate or GYY4137 before loading the cartridge into the XF96 instrument. Spent GYY4137 was used as a control compound. GYY4137 (final concentrations of 0, 10, 25, 50 and 100 μM) or DTT (final concentrations of 0, 1.25, 2.5, 5 and 10 mM) were also introduced through the drug ports in the XF96 plate by serial injection. Since GYY4137-generated H$_2$S bleeds over to adjacent wells, control wells filled with medium were used to separate experimental wells. Untreated control cells received either distilled water (UT) or spent GYY4137 (CT). Heat-killed *Mtb* cells were used as additional controls to rule out enzyme-mediated changes in OCR.

**Bioenergetic analysis using Oroboros Oxygraph-2k.** The OCR of *M. bovis* BCG (nonvirulent mycobacterium genetically related to *Mtb*) was measured using an Oroboros Oxygraph-2k respirometer (Oroborus Instruments, Austria). *M. bovis* BCG cells were grown as outlined above and placed in the sample chamber. Freshly prepared NaHS was administered through the injection ports and readings were recorded.

**Total ATP quantitation in mycobacterial cells.** Total ATP was quantified with the bioluminescence-based Enliten ATP assay system (Promega, USA) as per the manufacturer's instructions. Briefly, 100 μl aliquots of bacterial culture were collected at designated times and immediately heat inactivated and cell debris pelleted by centrifugation as described[79]. Twenty-five microliter aliquots of cleared cell lysate were transferred into 96-well assay plates and analyzed according to the manufacturer's instructions. Luminescence was detected using the luminescence mode in a Synergy Hybrid plate reader (Biotek Instruments, USA) and was expressed as relative luminescence units (RLU). Readings were normalized against CFU counts.

**ROS production in the presence of H$_2$S.** *Mtb* H37Rv was grown in MB 7H9 media supplemented with 0.2% glycerol, 1× ADS (albumin, dextrose and saline) and 0.01% tyloxapol at 37 °C. ROS production in *Mtb* was measured using dihydroethidium (DHE; Excitation/Emission of 500/605 nm). *Mtb* cells were filtered through a 10 μm filter to generate a single-cell suspension, and treated with GYY4137 (0, 10, 25, 50 μM) for 1 h. Cumene hydroperoxide at 15 mM was used as positive control. DHE (10 μM) was added to the treated bacilli and incubated for a further 30 min at 37 °C. Prior to analysis, treated bacilli were washed once using fresh media to remove any residual extracellular DHE. The fluorescence of DHE stained samples was acquired using a FACS Aria III cell sorter (488 nm laser excitation, and BP 610/20 nm for emission acquisition). Cells were acquired at a constant flow rate (setting 4) and the threshold rate of approximately 1000−2000 events per second; 100,000 total events were recorded per sample. The bacterial population was identified according to the forward and side scattering property of the population (FSC versus SSC). The doublet population of bacteria was discriminated from the analysis according to the relationship of the width and area of the SSC signal pulses of *Mtb*. Mean fluorescent intensity was calculated using FlowJo software package.

**LC-MS/MS targeted metabolic analysis.** Intracellular ATP, NADH, NAD$^+$, and metabolite levels were measured as described elsewhere[30,80]. *Mtb* H37Rv cultures (OD$_{600}$ ~ 0.5−0.6) grown in MB 7H9 broth supplemented with 0.2% glucose, 0.08% NaCl, 0.5% BSA and 0.02% tyloxapol were exposed to GYY4137 under carefully controlled conditions to final concentrations of 10, 25 and 100 μM. At 1 and 24 h after exposure, the OD$_{600}$ of each culture was recorded and 1.8 ml of each culture was pelleted, 800 μl of cold chloroform: methanol at a ratio of 2:1 (containing the internal standards, 6-amininicotinic acid and mono-methyl fumarate) was added and culture lysates were stored at −80 °C until further processing. The bacterial lysate in the chloroform:methanol mixture was filtered (0.22 μm filter) and the filtrate evaporated under reduced pressure during centrifugation. The dried pellet was reconstituted in 200 μl water overnight and submitted for LC-MS/MS analysis. All experiments were carried out with an Agilent 1200 series binary HPLC system (Agilent Technologies, Germany) coupled to an Applied Biosystems/MDS Sciex QTRAP™ 5500 linear accelerator trap mass spectrometer (AB/MDS Sciex, Canada) fitted with an ESI source. The mass spectrometer was operated in negative ion mode and samples analyzed using multi reaction monitoring (MRM). Data were acquired and analyzed using the Analyst (Version 1.6.2) software package (AB/MDS Sciex, Canada). Analytes were separated on a Waters Xterra C18 column (2.1 × 100 mm, 3.5 μm) using a gradient elution program at a flow rate of 200 μl/min and an injection volume of 5 μl. Two mobile phases were used: mobile phase A (A; 10 mM tributylamine, 15 mM acetic acid in 3% acetonitrile, 97% water) and mobile phase B (B; 10 mM tributylamine, 15 mM acetic acid in 100% acetonitrile). The gradient program was as follows: 0−1 min, 0% of A; 1−10 min, a linear increase from 0 to 100% of B; 10−15 min, 100% of B; 15−15.1 min, 0% of B; and 15.1−25 min, 0% of B. The AUC were normalized to the OD$_{600}$ (10$^8$ bacilli/ml) of the sample cultures before ratios were determined.

**Fractional nutrient contribution (FNC).** *Mtb* H37Rv was cultured in MB 7H9 media (Difco) supplemented with 10% OADC (Difco) and 0.01% Tyloxapol (Sigma-Aldrich) at 37 °C. All $^{13}$C-tracer carbon sources ([$^{13}$C$_6$] glucose and [$^{13}$C$_2$] acetate) were purchased from Sigma-Aldrich. *Mtb* cultures were grown till an OD600 ~0.6, pelleted and washed twice using MB 7H9 media (Difco) containing 0.01% Tyloxapol. After the final centrifugation step, the pellets were resuspended in 4.5 ml experiment media, MB 7H9 containing the one $^{13}$C-tracer carbon source (final concentration 0.2%) and GYY4137 to a final concentration of 0, 10, 25, 50 or 100 μM respectively. Cultures were grown incubated overnight (~16 h) at 37 °C, samples (4 ml) were centrifuged and pellets were kept on dry ice until further processing. The frozen pellets were resuspended 1.8 ml extraction buffer (2:2:1 methanol: acetonitrile:water) and bacilli lysed via five rounds of bead beating (7000 rpm for 60 s). Cell lysates were sterilized using 0.2 μm membrane spin filters. Protein concentrations of bacterial lysates were measured using a Micro BCA™ Protein Assay Kit (Thermo Scientific). Cell lysates were dried under reduced pressure and the resultant pellets were re-dissolved in water and analyzed for organic acid analysis by LC-MS. For amino acid analysis by LC-MS, the re-dissolved cell lysates were diluted with acetonitrile (50% final concentration) before analysis.

Organic acids were separated on a Biorad Aminex HPX-87 column (300 × 7.8 mm), using an aqueous 0.1% formic acid isocratic mobile phase program at a flow rate of 400 μl/min and an injection volume of 20 μl. Amino acids were separated on a Waters BEH Amide column (2.1 × 100 mm, 1.7 μm) using a gradient elution program at a flow rate of 200 μl/min and an injection volume of 1 μl. Two mobile phases were used: mobile phase A (A; aqueous 0.1% formic acid) and mobile phase B (B; acetonitrile, 0.1% formic acid). The gradient program was as follows: 0–0.2 min, 99% of B; 0.2–24 min, a curvilinear decrease from 99 to 30% of B; 24–25 min, 30% of B; 25.1–35 min, 99% of B. Both columns were connected to a Dionex Ultimate 3000 UPLC, and the total negative (organic acids) and positive (for amino acids) ion chromatograms (50–750 m/z scan range) were collected using a Thermo Scientific Q Exactive mass spectrometer. The total ion chromatograms of all relevant metabolites, and their respective isotopologues, were visualized and analyzed using Skyline V 3.7 (freely available open source software developed by MacCoss Lab, University of Washington, Seattle, WA USA). Chromatographic data were analyzed using Windows Excel 2016 and statistical analysis and data representation performed using GraphPad Prism 7.02.

FNC was calculated as previously reported[33]. For a given metabolite, the fraction of $^{13}C$ isotopologues in a specific metabolite pool is calculated as the sum of all the labeled isotopologues divided by the total abundance of the specific metabolite (all labeled and unlabeled isotopologues). For instance, in the case of glucose/fructose-6-phosphate, FNC = [(M + 1) + ⋯ + (M + 6)]/[(M + 0) + ⋯ + (M + 6)]. Thus, independent culturing of $Mtb$ using $^{13}C_6$-glucose and $^{13}C_2$-acetate as carbon sources ($^{13}C$-tracers) yielded FNC values of 0.463 and 0.016 respectively (Fig. 3). The FNC of "other" carbon sources utilized by the bacteria is subsequently calculated by $FNC_{other} = 1 − (FNC_{glucose} + FNC_{acetate}) = 0.548$. The same method was used to calculate the FNC for the "Other" metabolite pools. Glutamate is a likely source of $FNC_{other}$ as MB 7H9 medium used in these experiments contains a substantial amount of glutamate (0.5 g/l).

**Mycobacterial growth and ATP quantitation with •NO donor.** The •NO donor DETA/NO (Sigma, USA) was freshly prepared according to the manufacturer's instructions. Cultures (10 ml) receiving either DETA/NO or GYY4137 alone were sealed in sterile 60 ml PETG bottles following addition of reagents. For cultures receiving both DETA/NO (250 μM) and GYY4137, GYY4137 was added 15 min after the addition of DETA/NO and the cultures immediately sealed. For each time point, aliquots were removed to perform ATP assays and CFU analysis. Cultures were plated and CFU enumerated as described above. ATP analysis was performed as described in the earlier section.

**Nitrate/nitrite estimation.** Nitrate/nitrite estimation was performed on mouse sera at 8 weeks post infection, or on culture media of infected macrophages using a nitrate/nitrite fluorometric assay kit (Cayman Chemicals, USA) according to the manufacturer's instructions. Mice used for this experiment were age-matched ($n = 4–5$).

**Methylene blue method for sulfide measurement.** The concentration of $H_2S$ (determined as a combination of free $H_2S$, $HS^−$ and $S^{2−}$) in serum was measured spectrophotometrically using the methylene blue method. Briefly, serum/cultures were mixed with 0.25 ml Zn acetate (1% w/v) and 0.45 ml water for 10 min at room temperature. Trichloroacetic acid (TCA) (10%; 0.25 μl) was then added and centrifuged (14,000 × g; 10 min; 4 °C). The supernatant was collected and mixed with $N$, $N$-dimethyl-p-phenylenediamine sulfate (20 μM) in 1.2 M HCl and $FeCl_3$ (30 μM) in 1.2 M HCl. Absorbance was subsequently monitored at 670 nm and the concentration of $H_2S$ (defined as above) was determined using a standard curve of freshly prepared NaHS (0–200 μM; $R^2 = 0.9987$).

**Detection of $H_2S$ using the fluorescent probe WSP-1.** WSP-1 (Cayman Chemicals, USA) is a reactive fluorescent probe designed to detect $H_2S$ in biological samples and cells, and rapidly reacts with this gas to generate benzodithiolone and a fluorophore with excitation and emission maxima 465 and 515 nm, respectively. WSP-1 was prepared according to the manufacturer's instructions.

**Hypoxia studies in mycobacteria.** For hypoxia studies, 30 μl of 100 mg/ml metronidazole (MN) was added to 30 ml of actively growing bacterial culture in 60 ml PETG bottles followed by addition of 25 μM GYY4137 or spent GYY4137, sealed and kept at 37 °C without agitation. Control $Mtb$ cultures from the same inoculum were processed similarly without MN or GYY4137. For prolonged hypoxia experiments, cells were processed using a well-established hypoxia model in $Mtb$[81]. All the cultures received either 25 μM GYY4134 or spent GYY4137 just prior to sealing the culture bottles. Following hypoxia reactivation studies, post-hypoxia cultures were pelleted, washed and resuspended in fresh media. Equal numbers of cells were inoculated in fresh media and monitored by absorbance ($OD_{600}$) and CFU plating of culture aliquots at different time points. Colonies were enumerated after 4−6 weeks of incubation at 37 °C.

**Oxidative stress assays.** $Mtb$ cells were grown to 0.6−0.8 $OD_{600}$ and exposed to 25 μM GYY4137 for 6 h. $2 × 10^6$ cells were transferred into each well of a 96-well plate[29]. The oxidative stress agents menadione (MND, 0.5 mM) or cumene

hydrogen peroxide (CHP, 0.5 mM) were added to the cells followed by incubation at 37 °C for 6 h. Unexposed cells received only spent GYY4137 and were processed in a separate plate. Cells were subsequently plated, and colonies were enumerated after 4−6 weeks. The percentage of surviving cells was determined relative to untreated cells.

**Alamar blue assays.** $Mtb$ (200 μl of cells; $OD_{600} = 0.4$) was incubated with various concentrations of aminooxyacetic acid (AOAA) or cysteine for 18 h, followed by incubation with Alamar blue for 24 h. Absorbance was then read at 570/600 nm. Control wells contained growth medium and AOAA or cysteine (at each respective concentration) as well as Alamar blue.

**Isolation of RNA and sequencing.** $Mtb$ cultures were exposed to 25 μM GYY4137 or spent GYY4137 for 1 h under carefully controlled conditions ($n = 3$ per group). Total RNA was isolated from $Mtb$ (0.6−0.8 $OD_{600}$) using Trizol as described previously[29]. RNA purity was determined by measuring the ratio of UV absorbance at 260 and 280 nm. A parallel set of culture vials containing methylene blue was used as an indicator to ensure that hypoxia was not generated under these conditions. Ribosomal RNA depletion was performed using a Ribo-Zero kit (Illumina, USA) and the integrity of the RNA was assessed using an Agilent 2100 Bioanalyzer. mRNA sequencing was performed using an Illumina HiSeq2500 with the latest sequencing reagents and flow cells providing up to 300 Gb of sequence information per cell. Briefly, cDNA libraries were prepared using TruSeq library generation kits (Illumina USA). Library construction consisted of random fragmentation of the mRNA, followed by cDNA production using random primers. The cDNA libraries were quantitated using qPCR in a Roche LightCycler 480 with a Kapa Biosystems kit for library quantitation (Kapa Biosystems, USA) prior to cluster generation. Clusters were generated to yield approximately 725−825K clusters/mm². For each sequencing run, >10 million paired end ($2 × 50$ bp) reads were generated and the raw RNA-Seq fastq reads were aligned to the $Mtb$ H37Rv genome using the short-read aligner Bowtie2 version 2.2.3. EDGE-Pro (Estimated Degree of Gene Expression in PROkaryotes) version 1.3.1 was used to estimate gene expression levels. Finally, significance and differential transcript expression was ascertained using DESeq2 version 1.10.1. Genes meeting the desired criteria (fold change ≥ ± 1.50 and $q$ value < 0.05) were considered differentially regulated.

**Isolation of RNA and quantitative RT-PCR.** All mycobacterial strains ($Mtb$ H37Rv, $\Delta dosS$, $\Delta dosT$ and $\Delta dosST$) were grown to mid-log phase. Cultures received 25 μM GYY4137 or spent GYY4137 for 1 h and were processed under carefully controlled conditions similar to the cultures processed for RNA seq analysis. Total RNA was isolated using RNApro (MP Biomedicals, USA) according to the manufacturer's instructions. Extracted RNA was further treated with DNase (Thermo Scientific, USA) to remove any trace of genomic DNA. The purity of the RNA was determined by measuring the ratio of UV absorbance at 260 and 280 nm. One hundred nanograms of total RNA was used for cDNA synthesis using the iScript cDNA synthesis kit (Bio-Rad, USA). Quantitative RT-PCR was performed using SsoAdvanced SYBR green supermix (Bio-Rad, USA) with the Bio-Rad CFX96 detection system according to the manufacturer's instructions. Quantitative RT-PCR reactions were performed in duplicate using three independent biological replicates. Relative fold changes in gene expression were determined using the $2\Delta C_t$ method[82], where $C_t$ values of target genes were normalized to $C_t$ values of $Mtb$ 16sRNA as an internal control. Relative fold change was determined as the ratio of expression of respective genes between untreated and GYY4137-treated cultures. Primers used in this study are listed in Supplementary Table 1.

**Statistical analysis.** All measurements were taken from distinct samples. Statistical computations were performed with Prism 6.01 software (GraphPad, La Jolla, California). Pairwise comparisons were performed by parametric unpaired $t$ test. Multiple comparisons were performed using one-way or two-way ANOVA followed by Tukey's or Sidak's multiple comparisons tests. The log-rank (Mantel–Cox) test was used to determine median survival time.

## Data availability

RNA-Seq gene expression data have been deposited in the EMBL-EBI ArrayExpress database under accession E-MTAB-7421. The source data underlying Figs. 1–6 are provided as a Source Data file.

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

## Acknowledgements

This work was supported by NIH grants R01AI111940, R01AI134810, R21A127182 and a Bill and Melinda Gates Foundation Award (OPP1130017) (A.J.C.S.), R01HL092857 (S.M.B.), P30DK079626 (UAB DRC BARB Core) and T32HL007918 (A.S.), by pilot funds from the UAB CFAR, CFRB and Infectious Diseases and Global Health and Vaccines Initiative, and the CRDF Global to A.J.C.S. The research was also co-funded by the South African Medical Research Council to A.J.C.S. V.S. was supported by American Lung Association fellowship RT-232840-N, a UAB CFAR Development grant, and Innovative Young Biotechnologist Award BT/11/IYBA/2018/01 (Department of Biotechnology, Government of India). The authors wish to thank the staff and management of Southeastern Biosafety Laboratory Alabama Birmingham (SEBLAB), an NIAID-supported (UC6 AI058599) regional biocontainment laboratory.

## Author contributions

Conceptualization and design: V.S., A.J.C.S. Execution: animal survival studies (V.S., J.N.G.), organ burden experiments (V.S., K.C.C., V.P.R., J.N.G.), cytokine analysis (V.S., V.P.R.), FACS analysis (V.S., K.C.C., V.P.R.), in vitro growth and biochemical assays (V.S.), XF assays (V.S., D.A.L., J.S.M.), LC-MS (D.A.L., J.H.A., M.A.R.), O2-K experiments (V.S., A.S., D.R.M.), western blot (K.C.C.), RNA Seq analysis (V.S.), RNA preps and isolation (V.S., V.P.R.), qRT-PCR validation (K.C.C.), in vitro stress assays (V.S., V.P.R., M.A.R., T.T.R.K.), macrophage assays (V.S.), animal drug studies: (V.P.R., V.S.), cloning and *cydC* complementation (B.E.T.). Data analysis: V.S., J.R.L., J.N.G., K.C.C., V.P.R., D.R.M. and A.J.C.S. Technical expertise: S.M.B., D.R.M., T.T.R.K., and J.R.L. Writing: V.S., J.N.G. and A.J.C.S. Editing: J.R.L., J.N.G., V.P.R. and K.C.C. Figure preparation: V.S., V.P.R. and K.C.C. All authors discussed the results and commented on the manuscript.

## Competing interests

Authors declare no competing interests.
