## [Peer Review File · Nature Communications]

Reviewers' comments:

Reviewer #1 (Remarks to the Author):

This is a very interesting manuscript that explores the role of hydrogen sulfide in the growth of an intracellular pathogen in an animal model. The group demonstrates that a mouse that's deficient in one copy of the *Cbs* gene, that regulates cystathione-beta synthase, and ultimately sulfur metabolism in mice leads to prolonged survival of this animal model. This work then suggests that hydrogen sulfide plays a role in stimulating the regulation of respiration of *Mtb* cells and ultimately may play a role into the persistence phenomenon of *Mtb*. The uniqueness of this paper is that they combine respirometry analyses with metabolomic and transcriptomics to characterize how *Mtb* responds to differences in hydrogen sulfide concentrations. The authors convincingly show that hydrogen sulfide reverses NO inhibition of *Mtb* respiration. This appears to be mediated through the cytochrome bd oxidase. To my knowledge this is the first report that hydrogen sulfide regulates the growth of any pathogen and thus, is likely to be of interest for broad readership.

Minor Point: The paper is very well written but they should change description of the transposon insertion in figure S4 and in the accompanying figure legend. They describe the transposon insertion in the *cydC* gene as Tn::*cydC*. Following guidelines for bacterial mutations it should be written as *cydC*::Tn as this reflects a transposon insertion within a *cydC* gene. Ideally they should give it an allele number following the name of the transposon so it can be compared in future studies to other transposon insertions within these genes or deletion mutants made within the gene.

Reviewer #2 (Remarks to the Author):

The manuscript by Saini et al. describes the role of hydrogen sulfide in the pathogenesis of *Mycobacterium tuberculosis*. The paper reports that hydrogen sulfide enhances bacterial growth in vitro, survival in host macrophages, and pathogenicity in a mouse infection model. Hydrogen sulfide also stimulates respiratory chain activity and causes a remodeling of central metabolic pathways. The authors use a variety of model systems and a broad array of techniques. The experiments seem to be carefully done and the paper generally is well written.

The manuscript does not fully clarify the mechanism of hydrogen sulfide action, i.e. how this molecule manipulates the host cell to produce more hydrogen sulfide and how exactly it enhances respiratory activity. Nevertheless, this reviewer expects that the manuscript will have a strong impact in the field.

Please address the following minor comments:

1. The authors regard cytochrome bd as the primary target of hydrogen sulfide (page 18, line 453). Is it really the main target or is *bc1/aa3* the main target (as reported for the respiratory chain in *E. coli*) and cytochrome bd mediates the enhanced respiratory flux that is triggered by inhibition of *bc1/aa3*?
2. Related to the previous point: if hydrogen sulfide indeed interferes with the terminal oxidases of the respiratory chain and its effect on *bc1/aa3* is stronger than on cytochrome bd, then upregulation of cytochrome bd can be expected, as earlier shown for inhibition of ATP synthase and inhibition/inactivation of *bc1* (Refs 34, 38, 59). Did the RNAseq experiments indicate such an upregulation?
3. In Figure 5 the authors investigate the *dos* regulon in response to hydrogen sulfide. Mutant strains show clearly less expression of the selected *dos* regulon genes, indicating a role of *dosS* and *dosT*. In the main text (page 12, line 314) the authors argue that their results are consistent

with regulation of the dos regulon by the copper regulator CsoR. What is the relation between CsoR and dosS or dosT? Do the authors mean that hydrogen sulfide stimulates CsoR, which subsequently induces the dos regulon via dosS or dosT? Please clarify.

4. Textual:

Figure 6: AOAA was used as inhibitor. Please state the full name of this inhibitor somewhere, best in figure legend.

Page 19, line 475: induction by "Cys". Please write in full.

Figure S8: The legend states that CydBD is functional only under hypoxic conditions. However, several reports (e.g. ref 29) indicate that this oxidase may be present and functional under normal oxygen tensions as well. Please rephrase. "XF96" is a specialist term. Please replace by "flux analysis" or similar.

Panel D of his figure states that 25 and 100 microM GYY4137 were used, the legend says 25 and 50 microM. Please clarify.

The y-axis of panel D and panel E reads "relative to GYY4137". What does that mean? Or are these values calculated relative to a sample where medium control was added?

Please check Fig S8 for accuracy.

Reviewer #3 (Remarks to the Author):

Based on an impressive amount of data collected on wild-type and CBS-deficient mice, macrophages and cultures of *Mycobacterium tuberculosis* (and other non-pathogenic mycobacterial strains), using a variety of complementary methodological approaches, including RNA-seq, metabolomics, protein immunodetermination, qRT-PCR, Seahorse bioenergetic profiling, high resolution respirometry and fluorimetric/colorimetric quantitation of selected metabolites, here it is claimed that Mtb exploits host CBS-derived H₂S to enhance growth and promote TB disease. The manuscript is well written and contains novel and highly interesting information which could have a high impact in the field. The effects of H₂S on Mtb growth, bioenergetics, metabolism and survival in mice are notable and well documented, whereas less convincing is at this stage the mechanism suggested to account for such effects. The following points need be addressed:

1) The involvement of cytochrome bd (encoded by *cydAB* genes) is claimed based on the data collected on the Tn:*cydC* mutant. This mutant is defective in cytochrome bd activity. However, the *cydC* gene codes not for cytochrome bd, but for an ABC transporter (CydDC) which was shown to export glutathione and cysteine in several bacteria (see the recent review by Poole et al. in press in *Research in Microbiology*). Mutation of this gene is thus expected to cause not only cytochrome bd deficiency, but also other important changes (as documented for other bacteria) which might be responsible for the observed effects of H₂S in Mtb. The finding that no differentially regulated genes are found in Tn:*cydC* cells exposed to GYY compared to unexposed Tn:*cydC* cells is important, but I wonder if there are genes which are differentially regulated in Tn:*cydC* cells compared to WT cells regardless of GYY. For the sake of this study, it would be important to assess the impact of the mutation beyond causing defective cytochrome bd assembly. In the absence of this information, the conclusion that cytochrome bd plays a major role in mediating H₂S effects should be toned down, unless it is shown that the same results obtained with the *cydC* mutant can be achieved mutating *cydA* or *cydB*.

2) I am not convinced that the data in Fig. S10 rule out the involvement of SQR. Data in panel A show no effect of H₂S on O₂ consumption in either WT or the TN:rv0331 mutant. To rule out a contribution by SQR, I would have expected to see O₂ consumption to be enhanced by H₂S in both strains and to a similar extent. Also, in these experiments on WT or TN:rv0331 mutant did you attempt to administer H₂S as NaHS rather than GYY? What is known about mycobacterial SQRs?

3) The AOAA inhibitor interestingly reduced Mtb recovery from infected macrophages (Fig. 6I-J)

and from lungs of infected mice (Fig. 6K). To rule out possible direct effects of the inhibitor on Mtb, the effect of AOAA on Mtb growth should be tested. Related to this issue, it seems relevant to mention if Mtb codes for H₂S-synthesizing enzyme, orthologs of the mammalian ones or others (see Nzungize et al. (2019) *J Drug Target*. 27(9):1004-1016). Also, did you test if AOAA ameliorates lung morphology or survival of infected mice?

4) Data in Fig.5E showing that GYY makes Mtb more susceptible to metronidazole killing were interpreted as a clue for H₂S facilitating early entry of Mtb cells into hypoxic dormancy. Couldn't the higher susceptibility be just due to H₂S-induced up-regulation of ferredoxin, a well-known activator of metronidazole to its cytotoxic radical form?

5) Data in Fig.S6E-F are puzzling in that they show that mycobacterial growth followed over several days is enhanced by H₂S administered as a bolus of NaHS. Whereas GYY is a slow H₂S releaser, release of H₂S by NaHS is immediate and stability of H₂S in aqueous media is known to be rather poor (hours according to Nagy et al. (2014) *Biochimica et Biophysica Acta* 1840, 876–891). Then, how do you explain its persisting effect? Did you test the stability of H₂S under the experimental conditions used? Along the same lines, in Fig.6H it is shown that spermine NONOate affects mycobacterial growth at 8 and 24 hours after administration, despite its relatively short half-time (ca. 40 min at 37°C). The effect is attributed to reversible inhibition of respiration. However, in the presence of O₂, once NO in solution vanishes, respiration is typically restored quite quickly (minutes to tens of minutes), particularly at 37 °C. How come the effect of NO persists? These issues need to be discussed.

6) Some of the data obtained in this study seem in contrast to those reported for *M. smegmatis* in Garget et al. (2006) *J. Biol. Chem.* 281, 38712–38720. This needs to be discussed.

7) Reference to the literature can be improved. Apart from the suggestions below, I suggest quoting the following papers:

- Nambi et al. (2015) *Cell Host Microbe* 17, 829–837 (effect of H₂S on the antioxidant defence system and growth of Mtb mutants)
- Pal et al. (2018) *IUBMB Life*. (2018) 70(5):393-410 (review on H₂S and bacterial physiology/pathogenesis)

8) The manuscript architecture might be improved too. In the Results section, it seems a bit odd that data on the pharmacological inhibition of CBS in mice are presented last, after the experiments on Mtb. Please consider moving this subsection above after the subsection on mice survival. The Materials and Methods section could also be slightly rearranged moving the subsection on H₂S donors after that one on study design and grouping together as much as it is possible the subsections on macrophages or Mtb as distinct from those on mice.

Additional points:

- Several acronyms throughout the manuscript are not defined. Please define them all at their first use
- Line 72 ("A role for H₂S bacterial pathogenesis has not yet been described): please see Pal et al. (2018) *IUBMB Life*. (2018) 70(5):393-410
- Line 77: for cytochrome c oxidase inhibition by H₂S please consider quoting the pioneering papers by Lars Petersen (*Biochim Biophys Acta*. 1977 May 11;460(2):299-307) and Peter Nicholls (ref.58) instead of refs.14 and 15
- Line 78: ref 17 by Collman et al. seems mis-quoted here.
- Line 95: "peritoneal" instead of "intraperitoneal"
- Line 167 ("Unlike widely used NO and CO donors ..."): DETA NONOate is a slow donor too
- Line 258: CydBD can be misleading as an abbreviation for cytochrome bd. Please consider using the abbreviation CytBD or CydAB from the coding genes.
- Line 279: "small" instead of "marginal" (after all, the difference is statistically significant)
- Line 359: instead of quoting single specific papers (refs.49 and 50), please consider quoting

- some recent reviews on H₂S and NO crosstalk, such as Giuffrè & Vicente (2018) *Oxid Med Cell Longev.* 2018:6290931 and Cirino et al. (2017) *Br J Pharmacol.* 174(22):4021-4031
- Lines 456-458: the mechanism of sulfide metabolism by cytochrome c oxidase was revised in Nicholls et al. (2013) *Biochem Soc Trans.* 41(5):1312-6.
 - Line 542: NaHS from Alfa Aesar was used. Polysulfide contamination in commercially available sulfide salts is an issue in the field (see Nagy et al. (2014) *Biochimica et Biophysica Acta* 1840, 876-891). Did you test its purity?
 - Line 546: 10 mM for GYY4137 exceeds the reported solubility in aqueous media (ca.8 mM according to the manufacturer)
 - Lines 557-558 ("All experiments were approved ...). Already said above.
 - Line 581: the isolation procedure of murine peritoneal macrophages should be described
 - Line 625: "viability test" instead of "viable test"
 - Lines 715-718: these two sentences need to be revised
 - Line 730: curvilinear?
 - Line 775: "emission", not "admission"
 - Line 776: the reported maxima for the WSP-1 probe are 465 and 515 (not 550) nm. Please check
 - Line 796 ("in triplicates with at least 2 technical replicates"): do you mean biological replicates?
 - Line 798-814 ("Isolation of RNA and sequencing"): description of the RNA retrotranscription step is missing
 - Line 803 ("Ribosomal RNA reduction"): do you mean depletion?
 - Line 814: ≥ 1.50 -fold change
 - Line 816-817: the source of the *dosS*, *dosT* and *dosST* mutant strains should be reported
 - Line 821 ("The integrity of the RNA ..."): do you mean the purity?
 - Line 828 ("Gyy4137 alone (10uM)": 25 uM according to the legend to Fig.1 (see line 1076)
 - Line 849 ("Control mice received vehicle control (PBS)": if along with spent GYY, this should be reported
 - Line 1081: "following" should be deleted
 - Legend to Fig.S1: were murine peritoneal or RAW macrophages used in these experiments?
 - Figure S8D-E (y-axis labels): relative to spent GYY4137?
 - Legend to Fig.S8 (line 196): cytochrome bc1/aa3 (Complex II/IV)
 - Legend to Fig.S8 (line 203): 100uM (not 50uM) GYY4137
 - Legend to Fig.S9 (line 213, "Number in brackets: number of gene ..."): no brackets present. Also, those are percentages, not number of genes.
 - Fig.S10: why was CCCP used?
 - Legend to Fig.S10 ("Different concentrations of GYY4137"): only 25uM is shown

Specific Reviewer comments:

Reviewer #1:

This is a very interesting manuscript that explores the role of hydrogen sulfide in the growth of an intracellular pathogen in an animal model. The group demonstrates that a mouse that's deficient in one copy of the Cbs gene, that regulates cystathione-beta synthase, and ultimately sulfur metabolism in mice leads to prolonged survival of this animal model. This work then suggests that hydrogen sulfide plays a role in stimulating the regulation of respiration of Mtb cells and ultimately may play a role into the persistence phenomenon of Mtb. The uniqueness of this paper is that they combine respirometry analyses with metabolomic and transcriptomics to characterize how Mtb responds to differences in hydrogen sulfide concentrations. The authors convincingly show that hydrogen sulfide reverses NO inhibition of Mtb respiration. This appears to be mediated through the cytochrome bd oxidase. To my knowledge this is the first report that hydrogen sulfide regulates the growth of any pathogen and thus, is likely to be of interest for broad readership.

Minor Point: *The paper is very well written but they should change description of the transposon insertion in figure S4 and in the accompanying figure legend. They describe the transposon insertion in the cydC gene as Tn::cydC. Following guidelines for bacterial mutations it should be written as cydC::Tn as this reflects a transposon insertion within a cydC gene. Ideally they should give it an allele number following the name of the transposon so it can be compared in future studies to other transposon insertions within these genes or deletion mutants made within the gene.*

Response:

We appreciate the Reviewer's attention to detail. To ensure the correct citation of this strain, we reviewed the original publication (Shi et al., 2005), which revealed that the *aph* (kanamycin resistance) gene (not an *aph*-containing transposon) was inserted in the *Nhe* I site in *cydC* (*rv1620c*). We now correctly identify this strain as either *rv1620c::aph* or *cydC::aph* in the text and figures of the revised manuscript. Annotation of the corresponding complemented clone was also corrected.

Reviewer #2:

The manuscript by Saini et al. describes the role of hydrogen sulfide in the pathogenesis of Mycobacterium tuberculosis. The paper reports that hydrogen sulfide enhances bacterial growth in vitro, survival in host macrophages, and pathogenicity in a mouse infection model. Hydrogen sulfide also stimulates respiratory chain activity and causes a remodeling of central metabolic pathways. The authors use a variety of model systems and a broad array of techniques. The experiments seem to be carefully done and the paper generally is well written.

The manuscript does not fully clarify the mechanism of hydrogen sulfide action, i.e. how this molecule manipulates the host cell to produce more hydrogen sulfide and how exactly it enhances respiratory activity. Nevertheless, this reviewer expects that the manuscript will have a strong impact in the field.

Please address the following minor comments:

1. *The authors regard cytochrome bd as the primary target of hydrogen sulfide (page 18, line 453). Is it really the main target or is bc1/aa3 the main target (as reported for the respiratory chain in E. coli) and cytochrome bd mediates the enhanced respiratory flux that is triggered by inhibition of bc1/aa3?*

Response:

We thank the Reviewer for pointing this out and apologize for any confusion regarding our statement in **line 453** of the original manuscript. Our conclusion in **line 288** of the original manuscript states: *These data suggest that GYY4137 targets the aa3-type cytochrome c oxidase and CydBD, but that the aa3-type cytochrome c oxidase is more sensitive to H₂S.* In the revised manuscript, we have rephrased this statement in **lines 296-299 and 541-543** of the revised manuscript and clarified that inhibition of cytochrome *bc1/aa3* leads to rerouting of electrons through cytochrome BD, which compensates for inhibition of *bc1/aa3*; therefore, cytochrome *bc1/aa3* is the main target.

2. *Related to the previous point: if hydrogen sulfide indeed interferes with the terminal oxidases of the respiratory chain and its effect on bc1/aa3 is stronger than on cytochrome bd, then upregulation of cytochrome bd can be expected, as earlier shown for inhibition of ATP synthase and inhibition/inactivation of bc1 (Refs 34, 38, 59). Did the RNAseq experiments indicate such an upregulation?*

Response:

We observed no increases in expression of *cydA*, *cydB*, *cydC* or *cydD* when *Mtb* cells were exposed to 25 μ M GYY4137. This is not surprising as we speculated in our previous study (ref 29) that cytochrome *bd* requires an “activation signal” rather than its expression being upregulated.

3. In Figure 5 the authors investigate the *dos* regulon in response to hydrogen sulfide. Mutant strains show clearly less expression of the selected *dos* regulon genes, indicating a role of *dosS* and *dosT*. In the main text (page 12, line 314) the authors argue that their results are consistent with regulation of the *dos* regulon by the copper regulator *CsoR*. What is the relation between *CsoR* and *dosS* or *dosT*? Do the authors mean that hydrogen sulfide stimulates *CsoR*, which subsequently induces the *dos* regulon via *dosS* or *dosT*? Please clarify.

Response:

We thank the Reviewer and recognize that the relationship between *CsoR* and the *Dos* regulon was not presented clearly. In the revised manuscript, we thoroughly discuss this issue in the **Results** section (a new figure panel [Figure 5D] is provided that summarizes the RNAseq data) and in the **Discussion** (lines 526-533).

4. Textual:

Figure 6: AOAA was used as inhibitor. Please state the full name of this inhibitor somewhere, best in figure legend.

Response:

We apologize for this omission: AOAA = aminooxyacetic acid. The full name of the inhibitor has been stated in the text (line 401), the Figure 6 legend, and in the Materials and Methods of the revised manuscript.

Page 19, line 475: induction by “Cys”. Please write in full.

Response:

Cysteine has been written in full throughout the revised manuscript.

Figure S8: The legend states that *CydBD* is functional only under hypoxic conditions. However, several reports (e.g. ref 29) indicate that this oxidase may be present and functional under normal oxygen tensions as well.

Response:

The Reviewer is correct; *CydBD* is “functional” under hypoxic conditions but can be pharmacologically activated using an anti-TB drug such as Q203 to reroute electrons away from Complex III/IV to *CytBD*. Although it can be argued that the response to Q203 indicates that *CydBD* is also functional under normoxic conditions, this does not reflect the physiological response of the bacillus to hypoxia. Again, as indicated in 2 above, this is not surprising as we speculated in our previous study (ref 29) that cytochrome *bd* oxidase requires an “activation signal” rather than its expression being upregulated. Hence, we have removed “only” from the legend of Figure S8. This sentence now reads as follows: “*CytBD* is functional under hypoxic conditions, whereas...”

Figure S8: Please rephrase. “XF96” is a specialist term. Please replace by “flux analysis” or similar.

Response:

We agree and have replaced XF96 with Agilent Seahorse XF96 Extracellular Flux Analyzer in line 724 in the Methods section and the Figure S8 legend in the revised manuscript.

Figure S8: Panel D of his figure states that 25 and 100 microM GYY4137 were used, the legend says 25 and 50 microM. Please clarify.

Response:

We thank the Reviewer for identifying this discrepancy; 25 and 100 μ M GYY4137 were used. This has been corrected in the revised Figure S8 legend.

Figure S8: The y-axis of panel D and panel E reads “relative to GYY4137”. What does that mean? Or are these values calculated relative to a sample where medium control was added?

Response:

We apologize for the ambiguous labelling of the Y-axes. Untreated (media control) *Mtb* cells were compared relative to GYY4137-treated *Mtb* cells. This is now indicated on the Y-axis in **Figure S8D** and **S8E** in the revised manuscript.

Please check Fig S8 for accuracy.

Response:

We have corrected the legend as well as **Figure S8D** and Y-axis labels.

Reviewer #3:

Based on an impressive amount of data collected on wild-type and CBS-deficient mice, macrophages and cultures of Mycobacterium tuberculosis (and other non-pathogenic mycobacterial strains), using a variety of complementary methodological approaches, including RNA-seq, metabolomics, protein immunodetermination, qRT-PCR, Seahorse bioenergetic profiling, high resolution respirometry and fluorimetric/colorimetric quantitation of selected metabolites, here it is claimed that Mtb exploits host CBS-derived H₂S to enhance growth and promote TB disease. The manuscript is well written and contains novel and highly interesting information which could have a high impact in the field. The effects of H₂S on Mtb growth, bioenergetics, metabolism and survival in mice are notable and well documented, whereas less convincing is at this stage the mechanism suggested to account for such effects. The following points need be addressed:

Response:

We appreciate the Reviewer's positive comments and attention to detail.

1) *The involvement of cytochrome bd (encoded by cydAB genes) is claimed based on the data collected on the Tn:cydC mutant. This mutant is defective in cytochrome bd activity. However, the cydC gene codes not for cytochrome bd, but for an ABC transporter (CydDC) which was shown to export glutathione and cysteine in several bacteria (see the recent review by Poole et al. in press in Research in Microbiology). Mutation of this gene is thus expected to cause not only cytochrome bd deficiency, but also other important changes (as documented for other bacteria) which might be responsible for the observed effects of H₂S in Mtb. The finding that no differentially regulated genes are found in Tn:cydC cells exposed to GYY compared to unexposed Tn:cydC cells is important, but I wonder if there are genes which are differentially regulated in Tn:cydC cells compared to WT cells regardless of GYY. For the sake of this study, it would be important to assess the impact of the mutation beyond causing defective cytochrome bd assembly. In the absence of this information, the conclusion that cytochrome bd plays a major role in mediating H₂S effects should be toned down, unless it is shown that the same results obtained with the cydC mutant can be achieved mutating cydA or cydB.*

Response: We appreciate the Reviewer's concern that the *cydC* mutant may not confer a phenotype identical to that of a *cydAB* mutant, in light of studies in *E. coli* demonstrating that CydDC can also export cysteine and glutathione (Holyoake et al., 2016; Pittman et al., 2002).

To address to the Reviewer's comment, "For the sake of this study, it would be important to assess the impact of the mutation beyond causing defective cytochrome bd assembly", we have examined whether the *Mtb cydC* mutant is sensitive to cysteine concentrations exceeding physiological levels in the lung, and found that this is not the case. These data are now included in **Figure S9** of the revised manuscript.

Different from the Reviewer's statement above, this study is focused on respiration and bioenergetics of *Mtb* in response to host H₂S. We believe there is compelling evidence that the *cydDC* mutant is equivalent to the *cydAB* mutant for the reasons described below. Note that others (Poole et al., 2019) acknowledge that the *cydDC* mutants show all phenotypes of *cydAB* with additional defects. Therefore, in addition to our cysteine experiment, we believe the studies referenced below further address the Reviewer's concern ("...assess the impact of the mutation beyond causing defective cytochrome bd assembly.").

Firstly, this same issue was addressed by us in a recent study (Moosa et al., 2017) where we asked whether the *Mtb* genes coding for cytochrome *bd* oxidase are phenotypically equivalent, and whether disrupted oxidase function (*cydAB* disruption) can be separated from deficient ABC transport (*cydC* disruption). To do so, in-frame deletions of *Mtb cydA*, *cydAB* and *cydC::aph* (the same strain used in this study) were generated and compared in a series of functional assays. Importantly, we observed *identical* phenotypes for all of the *Mtb cyd* mutants (Moosa et al., 2017) and demonstrated that

disruption of either the CydAB oxidase or CydDC transport functions leads to an indistinguishable downstream effect.

Secondly, we examined *Mtb* respiratory function in the same series of *cyd* mutants (*cydA*, *cydAB*, *cydC*). All mutant strains displayed an identical bioenergetic phenotype (Moosa et al., 2017), which supports the findings of our prior study (Lamprecht et al., 2016) which used the same *Mtb cydC::aph* mutant used in this study.

Thirdly, while it may seem acceptable to functionally compare *E. coli cydDC* with *Mtb cydDC*, several factors could argue against this notion, for example:

- *E. coli cydAB* and *cydDC* code for cytochrome *bd* oxidase, but are separated from each other in the genome, whereas in *Mtb* these genes (*cydABDC*) are co-regulated and expressed in an operon. This suggests that the mode of regulation is distinct. Indeed, the *E. coli cydAB* operon is regulated by at least two global transcriptional regulators, Arc and Fnr (Cotter and Gunsalus, 1992), which are absent in *Mtb*. Based on these gene organization and regulatory factors, the functional roles of the cytochrome *bd* oxidases between these bacteria are likely to be different.
- *E. coli* contains two cytochrome BD oxidases (*bd-1* and *bd-2*) and one cytochrome *bo*₃ (Forte et al., 2016; Korshunov et al., 2016), and is a facultative anaerobe that resides in the gut and is exposed to mM quantities of H₂S. In contrast, *Mtb* is an obligate aerobe that resides primarily in the lung and contains only one cytochrome *bd* oxidase and a fused Complex III/IV oxidase.

With respect to the Reviewer's concern – “*In the absence of this information, the conclusion that cytochrome bd plays a major role in mediating H₂S effects should be toned down, unless it is shown that the same results obtained with the cydC mutant can be achieved mutating cydA or cydB*”, we respectfully submit that our previously published studies (Lamprecht et al., 2016; Moosa et al., 2017), additional new data, and other reasons highlighted above, clearly demonstrate that the *Mtb cydC* mutant exhibits the same phenotype as the *cydA*, *cydB* or *cydAB* mutants.

We thank the Reviewer for highlighting this issue, as it allowed us to more thoroughly describe the related findings in the revised manuscript.

2) I am not convinced that the data in Fig. S10 rule out the involvement of SQR. Data in panel A show no effect of H₂S on O₂ consumption in either WT or the TN:rv0331 mutant. To rule out a contribution by SQR, I would have expected to see O₂ consumption to be enhanced by H₂S in both strains and to a similar extent. Also, in these experiments on WT or TN:rv0331 mutant did you attempt to administer H₂S as NaHS rather than GYY? What is known about mycobacterial SQRs?

Response:

SQR codes for sulfide:quinone oxidoreductases, which are involved in sulfide detoxification *i.e.*, oxidation of H₂S and feeding of electrons directly into the electron transport chain. The Reviewer is correct that there is no effect of H₂S on oxygen consumption in both strains. Since H₂S stimulates *Mtb* respiration, disruption of *sqr* (*rv0331::aph*) should lead to an immediate reduction in respiration when H₂S is added, which is not observed. Therefore, we concluded that *rv0331* may either (i) not code for an SQR, or (ii) require specific conditions to be induced. The latter statement has now been included in the revised manuscript (**lines 495-496**).

Regarding the Reviewer's second question; we have used both NaHS and GYY4137. We prefer GYY4137 since addition of NaHS rapidly generates a bolus of H₂S, which quickly bleeds over into adjacent wells in the Agilent XF96 cartridge, thereby confounding the assay.

Regarding the Reviewer's third question regarding mycobacterial SQRs; we could not find any literature on mycobacterial or *Mtb* SQR activity, hence, not much is known about this gene.

3) The AOAA inhibitor interestingly reduced *Mtb* recovery from infected macrophages (Fig. 6I-J) and from lungs of infected mice (Fig. 6K). To rule out possible direct effects of the inhibitor on *Mtb*, the effect of AOAA on *Mtb* growth should be tested. Related to this issue, it seems relevant to mention if *Mtb* codes for H₂S-synthesizing enzyme, orthologs of the mammalian ones or others (see Nzungize *et al.* (2019) *J Drug Target*. 27(9):1004-1016). Also, did you test if AOAA ameliorates lung morphology or survival of infected mice?

Response:

We appreciate the Reviewer's concern. We wish to point out that that AOAA does not reduce *Mtb* burden in macrophages at 6 hr or 36 hr post-infection compared to initial *Mtb* burden (Figure 6I, light grey bars). In fact, CFUs are slightly increased at 36 hr in AOAA-treated macrophages compared to 0 h AOAA-treated controls.

Furthermore, our *in vivo* experiments (Figure 6K), demonstrate that the H₂S donor GYY4137 reverses the loss of *Mtb* organ burden observed in AOAA-treated wild type (*Cbs*^{+/+}) mice. This finding strongly indicates that it is the AOAA-mediated inhibition of CBS H₂S production, and not a direct effect of AOAA, that is the cause of reduced *Mtb* organ burden in AOAA-treated animals.

Nevertheless, to address the Reviewer's concern, we examined the direct effect of AOAA on *Mtb* proliferation *in vitro*. We observed that addition of AOAA directly to *Mtb* cultures at concentrations as high as 1 mM does not inhibit *Mtb* growth (Figure S12). Further, 2 mM AOAA inhibits ~35% of proliferating *Mtb* cells, suggesting that the actual concentration of AOAA in the phagolysosome of infected cells is significantly less than 2 mM. These data are discussed in lines 417-422 of the revised manuscript.

Overall, the macrophage and *in vivo* data suggest that AOAA does not kill *Mtb* in macrophages (Figure 6I) and certainly targets CBS, the effects of which can be fully reversed by GYY4137-derived H₂S (Figure 6K).

However, we would like to make the following comments. The only way to address in a scientifically rigorous manner if AOAA has any effects on *Mtb in vivo* or in macrophages, is to determine the AOAA concentration in the *Mtb*-containing phagolysosome and compare that to the *in vitro* concentration of AOAA that kills *Mtb*. This requires determining the volume of the infected phagolysosome to calculate the concentration (in molarity), which is technically unachievable for a variety of reasons. We are unaware of any anti-TB (or other) drug whose concentration has been determined in the phagolysosome.

Regarding the Reviewer's second comment "...relevant to mention if *Mtb* codes for H₂S-synthesizing enzyme..." Since this study is focused on the role of host-derived H₂S, we believe it would be distracting to speculate on H₂S-producing enzymes expressed in *Mtb*. Regarding the study by Nzungize, *et al.* 2019 mentioned by the Reviewer, we strongly question the methodology these authors used to support the conclusion that *Mtb metC* (*rv3340*) is a producer of H₂S. As one example, Figure 4 of this paper purports to show increased H₂S production by a recombinant *M. smegmatis* clone expressing *Mtb rv3340* as compared to WT *M. smegmatis*. However, increased H₂S production in the recombinant occurred only in the presence of NaHS (a H₂S donor), thereby making these results uninterpretable. Other data figures are equally problematic. In addition, respectfully, the paper is very poorly written, and does not meet basic standards for a peer-reviewed, published manuscript. Hence, we have not reference this paper.

Regarding the Reviewer's third question, we observed no detrimental effects of AOAA on mouse survival or behavior under our experimental conditions. Since the mice were treated for only 6 weeks, we did not perform a comprehensive histopathological analysis. It should be noted that AOAA is the best available pharmacological inhibitor of CBS (Asimakopoulou *et al.*, 2013; Hellmich *et al.*, 2015) and has been exhaustively tested in numerous animal studies (Hellmich *et al.*, 2015; Szabo, 2016; Szabo *et al.*, 2013), human xenografts (Hellmich *et al.*, 2015) and in humans (Perry *et al.*, 1980), yielding valuable safety and tolerability data.

4) Data in Fig.5E showing that GYY makes *Mtb* more susceptible to metronidazole killing were interpreted as a clue for H₂S facilitating early entry of *Mtb* cells into hypoxic dormancy. Couldn't the higher susceptibility be just due to H₂S-induced up-regulation of ferredoxin, a well-known activator of metronidazole to its cytotoxic radical form?

Response:

We agree with the Reviewer that any one or more of the ~5 ferredoxins (*rv0763c*, *rv1786*, *rv2007c* [*fdxA*], *rv1177* [*fdxC*], *rv3503c* [*fdxD*]) in *Mtb* could participate in the bioreductive activation of metronidazole.

Although only *rv2007c* (*fdxA*, a member of the Dos dormancy regulon) was found to be upregulated in our RNAseq analysis (**Figure 5C**), we have no evidence to suggest involvement of any of these proteins in the increased susceptibility to metronidazole. Rather, within the framework of our study, we believe the simplest explanation for metronidazole susceptibility is that H₂S stimulates oxygen consumption and promotes growth, thereby entering a hypoxic state faster compared to untreated cells.

5) Data in Fig.S6E-F are puzzling in that they show that mycobacterial growth followed over several days is enhanced by H₂S administered as a bolus of NaHS. Whereas GYY is a slow H₂S releaser, release of H₂S by NaHS is immediate and stability of H₂S in aqueous media is known to be rather poor (hours according to Nagy et al. (2014) *Biochimica et Biophysica Acta* 1840, 876–891). Then, how do you explain its persisting effect? Did you test the stability of H₂S under the experimental conditions used?

Response:

We understand the Reviewer's concern and thoroughly considered these issues (e.g. bolus effect of NaHS and the slow release property of GYY4137) prior to performing these experiments. Detailed experimental details are provided in the Materials and Methods. To specifically address the Reviewer's question regarding the persistent effect of H₂S in these assays, we ensured that *Mtb* was cultured in sealed vials. As a result, *Mtb* was continuously exposed to H₂S derived from GYY4137 or NaHS.

The Reviewer's second question regarding H₂S stability is unclear. H₂S is a weak acid and at physiological pH (~7.4) exists mainly as a hydrosulfide anion (HS⁻, 75-80%), with the remainder in the form of the dissolved uncharged H₂S gas (~20%) and negligible amounts of S₂⁻. Hence the sulfide pool (H₂S + HS⁻ + S₂⁻) is typically referred to as H₂S.

Along the same lines, in Fig.6H it is shown that spermine NONOate affects mycobacterial growth at 8 and 24 hours after administration, despite its relatively short half-time (ca. 40 min at 37°C). The effect is attributed to reversible inhibition of respiration. However, in the presence of O₂, once NO in solution vanishes, respiration is typically restored quite quickly (minutes to tens of minutes), particularly at 37 °C. How come the effect of NO persists? These issues need to be discussed.

Response:

We believe the Reviewer did not notice that we used diethylenetriamine NONOate (DETA/NO; 250 μM) in our experiments (**Figure 6**, legend), which has a half-life of 20 h (at 37° C, pH 7.4), and releases 2 moles of NO per mole of parent compound.

6) Some of the data obtained in this study seem in contrast to those reported for *M. smegmatis* in Garg, et al. (2006) *J. Biol. Chem.* 281, 38712–38720. This needs to be discussed.

Response:

The Reviewer is correct that our findings contrast with those of (Garg et al., 2006). This is not unexpected due to several major differences between the two studies, including, but not limited to:

- (Garg et al., 2006) infect their macrophages with the nonpathogenic, fast-growing *Mycobacterium smegmatis*, whereas we used highly pathogenic, slow-growing *Mycobacterium tuberculosis*
- (Garg et al., 2006) used healthy human monocyte-derived macrophages whereas we used mouse peritoneal macrophages deficient in CBS
- (Garg et al., 2006) used propargylglycine (PPG), an inhibitor of CSE, whereas we used AOAA, an inhibitor of CBS

Overall, we believe that there are too many differences between the study by Garg, et al. and this study to warrant further discussion as it will be distracting, resulting in an unfocused manuscript.

7) Reference to the literature can be improved. Apart from the suggestions below, I suggest quoting the following papers:

- Nambi et al. (2015) *Cell Host Microbe* 17, 829–837 (effect of H₂S on the antioxidant defence system and growth of *Mtb* mutants)

Response: We appreciate the Reviewer's suggestion. After a careful reading of the report by Nambi, *et al.*, it became clear that cysteine and NaHS (an H₂S donor) were used only as reagents to chemically complement a defect in a membrane-associated oxidoreductase complex (MRC), thereby stimulating mycothiol (MSH) synthesis and restoration of a normal MSH:MSSM ratio. This paper does not propose or consider a physiological role for H₂S in oxidant defense in *Mtb*. On this basis, we respectfully consider this report to be of little relevance to our manuscript and have not referenced it.

- Pal *et al.* (2018) *IUBMB Life*. (2018) 70(5):393-410 (review on H₂S and bacterial physiology/pathogenesis)

Response:

We agree with the Reviewer and have referenced this excellent review paper.

8) *The manuscript architecture might be improved too. In the Results section, it seems a bit odd that data on the pharmacological inhibition of CBS in mice are presented last, after the experiments on Mtb. Please consider moving this subsection above after the subsection on mice survival.*

Response:

We respectfully disagree with the Reviewer's suggestion regarding the order of data presentation. After careful consideration, it is clear that moving the animal data in Figures 6K and 6L and accompanying text to follow the subsection on mice survival, as suggested by the Reviewer, would require addition of another figure. Further, our data showing the impact of AOAA-mediated inhibition of CBS activity on CFUs (*in vitro* [Figure 6I] and *in vivo* (Figure 6K) and on serum sulfide levels (Figure 6L) are purposely grouped together.

The Materials and Methods section could also be slightly rearranged moving the subsection on H₂S donors after that one on study design and grouping together as much as it is possible the subsections on macrophages or Mtb as distinct from those on mice.

Response:

We agree with the Reviewer and have re-arranged the relevant sub-sections of Materials and Methods in the revised manuscript.

Additional points:

- Several acronyms throughout the manuscript are not defined. Please define them all at their first use

- Line 72 ("A role for H₂S bacterial pathogenesis has not yet been described): please see Pal *et al.* (2018) *IUBMB Life*. (2018) 70(5):393-410

Response:

All acronyms have been defined in the revised manuscript.

Regarding line 72, we are referring specifically to host-generated H₂S and not bacterial H₂S. This is now been clarified in the revised manuscript (line 73).

- Line 77: for cytochrome c oxidase inhibition by H₂S please consider quoting the pioneering papers by Lars Petersen (*Biochim Biophys Acta*. 1977 May 11;460(2):299-307) and Peter Nicholls (1982, ref.58) instead of refs. 14 and 15

Response:

We agree and have added the Petersen, *et al.* (1977) reference to the **Introduction** of the revised manuscript (line 77).

- Line 78: ref 17 by Collman *et al.* seems mis-quoted here.

Response:

We agree and have removed this reference from the revised manuscript.

- Line 95: "peritoneal" instead of "intraperitoneal"

Response:

We agree and have made this correction in the revised manuscript.

- Line 167 ("Unlike widely used NO and CO donors ..."): DETA NONOate is a slow donor too

Response:

We agree with the Reviewer; this NO donor has a half-life of 20 hours at 37°C at pH 7.4 and could be regarded as a "slow" releaser. However, GYY4137 releases H₂S over a period of several weeks. We have modified the text to read "unusually slow" (line 171), which is the point we wanted to make.

- Line 258: *CydBD* can be misleading as an abbreviation for cytochrome *bd*. Please consider using the abbreviation *CytBD* or *CydAB* from the coding genes.

Response:

We agree and have made the correction to *CytBD* throughout the revised manuscript.

- Line 279: “small” instead of “marginal” (after all, the difference is statistically significant)

Response:

We agree and have made this change in the revised manuscript.

- Line 359: instead of quoting single specific papers (refs.49 and 50), please consider quoting some recent reviews on H₂S and NO crosstalk, such as Giuffrè & Vicente (2018) *Oxid Med Cell Longev.* 2018:6290931 and Cirino et al. (2017) *Br J Pharmacol.* 174(22):4021-4031

Response:

We agree and have included these two references in the revised manuscript.

- Lines 456-458: the mechanism of sulfide metabolism by cytochrome *c* oxidase was revised in Nicholls et al. (2013) *Biochem Soc Trans.* 41(5):1312-6.

Response:

We thank the Reviewer for pointing this out. We have added this reference to the revised manuscript.

- Line 542: NaHS from Alfa Aesar was used. Polysulfide contamination in commercially available sulfide salts is an issue in the field (see Nagy et al. (2014) *Biochimica et Biophysica Acta* 1840, 876–891). Did you test its purity?

Response:

We understand the reviewer's concern regarding polysulfide contamination in commercially available sulfide salts and are well aware of this issue. This is a practical problem resulting from the natural susceptibility of sulfide to aerobic oxidation, and makes elimination of all polysulfide virtually impossible. Indeed, even H₂S gas, the purest source of H₂S, can produce polysulfide once in the solution, highlighting the inevitability of polysulfide formation while working with H₂S (Greiner et al., 2013). Importantly, the field has responded to these challenges, and clear recommendations have been made to prevent or minimize polysulfide formation (Hughes et al., 2009), which we have carefully followed in our experiments. This includes using the recommended salt for making H₂S (*i.e.* anhydrous sodium hydrogen sulfide from Alfa Aesar), which is not hygroscopic and remains white over several months in a vacuum desiccator (Hughes et al., 2009). As per these recommendations, our H₂S stock solutions contain metal chelators and are deoxygenated prior to use. Nonetheless, the possibility of having trace amounts of sulfide oxidation products, including polysulfide, cannot be completely ruled out.

- Line 546: 10 mM for GYY4137 exceeds the reported solubility in aqueous media (ca.8 mM according to the manufacturer)

Response:

We respectfully disagree with the Reviewer. The manufacturer (Cayman Chemical) states the solubility of GYY4138 to be ~8 mM in PBS, pH 7.2. Our 10 mM GYY4137 stocks were prepared in ultrapure water where GYY4137 solubility exceeds 75 mM as originally reported (Li, *et al.*, *Circulation*, 2008, 117: 2351-60).

- Lines 557-558 (“All experiments were approved ...). Already said above.

Response:

The Reviewer is correct. We have removed this sentence from the revised manuscript.

- Line 581: the isolation procedure of murine peritoneal macrophages should be described

Response:

We thank the Reviewer for pointing this out; a full description is now given in the Materials and Methods of the revised manuscript.

- Line 625: “viability test” instead of “viable test”

Response:

When preparing cells for flow cytometry, a fixable viability dye is used to discriminate between live and dead cells. In the revised manuscript, we have corrected the text to read, “viability dye”.

- *Lines 715-718: these two sentences need to be revised*

Response:

We agree and have corrected these sentences in the revised manuscript.

- *Line 730: curvilinear?*

Response:

Curvilinear is a standard mass spectrometry term used to describe non-linear gradients.

- *Line 775: “emission”, not “admission”*

Response:

We thank the Reviewer and have made this correction in the revised manuscript.

- *Line 776: the reported maxima for the WSP-1 probe are 465 and 515 (not 550) nm. Please check*

Response:

We thank the Reviewer for pointing this out and have corrected the wavelength to 515 nm in the revised manuscript.

- *Line 796 (“in triplicates with at least 2 technical replicates”): do you mean biological replicates?*

Response:

We apologize for any confusion. This line was included in the Methods section in error and has been omitted from the revised manuscript. Relevant details are in the **Figure 5** legend of the revised manuscript.

- *Line 798-814 (“Isolation of RNA and sequencing”): description of the RNA retrotranscription step is missing*

Response:

As requested, we have added additional experimental details to this section of the Methods.

- *Line 803 (“Ribosomal RNA reduction”): do you mean depletion?*

Response: We apologize for any confusion. The terms “ribosomal reduction” and “ribosomal depletion” are used interchangeably, as this process reduces ribosomal RNA by ~99%. We have changed the revised manuscript to read “depletion” for clarity.

- *Line 814: ≥ 1.50 -fold change*

Response:

We thank the Reviewer for pointing this out and have corrected this sentence in the revised manuscript.

- *Line 816-817: the source of the dosS, dosT and dosST mutant strains should be reported*

Response:

We thank the Reviewer for pointing this out and have included the source of these mutant strains in the revised manuscript.

- *Line 821 (“The integrity of the RNA ...”): do you mean the purity?*

Response:

The Reviewer is correct. “Purity” is the correct term and this error has been corrected in the revised manuscript.

- *Line 828 (“Gyy4137 alone (10uM)”): 25 uM according to the legend to Fig.1 (see line 1076)*

Response:

We thank the Reviewer. This error has been corrected to 25 μ M in the revised manuscript.

- *Line 849 (“Control mice received vehicle control (PBS)”): if along with spent GYY, this should be reported*

Response:

As stated, the vehicle control mice in this setting received PBS only without spent GYY4137.

- Line 1081: "following" should be deleted

Response:

We thank the Reviewer and have made this correction in the revised manuscript

- Legend to Fig.S1: were murine peritoneal or RAW macrophages used in these experiments?

Response:

Peritoneal macrophages were used in **Figure S1**. We have clarified this in the **Results** section and the **Figure S1** legend in the revised manuscript.

- Figure S8D-E (y-axis labels): relative to spent GYY4137?

Response:

We apologize for the confusion. We have corrected the Y-axis labels.

- Legend to Fig.S8 (line 196): cytochrome bc1/aa3 (Complex II/IV)

Response:

We thank the Reviewer for pointing this out and have corrected this in the revised manuscript.

- Legend to Fig.S8 (line 203): 100uM (not 50uM) GYY4137

Response:

We thank the Reviewer for catching this error and have corrected this in the revised manuscript.

- Legend to Fig.S9 (line 213, "Number in brackets: number of gene ..."): no brackets present. Also, those are percentages, not number of genes.

Response:

We apologize for the confusion; the dark colors masked the numbers in brackets. The colors and font have been changed for improved clarity in the revised manuscript.

- Fig.S10: why was CCCP used?

Response:

CCCP was used as an uncoupler to assess whether the membrane potential of the *Mtb* mutant strain collapses when exposed to GYY4137.

- Legend to Fig.S10 ("Different concentrations of GYY4137"): only 25uM is shown

Response:

We apologize for the confusion; only 25 μ M GYY4137 was used as this concentration stimulates OCR. This has been corrected in the revised manuscript.

REFERENCES

Asimakopoulou, A., Panopoulos, P., Chasapis, C.T., Coletta, C., Zhou, Z., Cirino, G., Giannis, A., Szabo, C., Spyroulias, G.A., and Papapetropoulos, A. (2013). Selectivity of commonly used pharmacological inhibitors for cystathionine beta synthase (CBS) and cystathionine- γ -lyase (CSE). *British journal of pharmacology* *169*, 922-932.

Cotter, P.A., and Gunsalus, R.P. (1992). Contribution of the *fnr* and *arcA* gene products in coordinate regulation of cytochrome o and d oxidase (*cyoABCDE* and *cydAB*) genes in *Escherichia coli*. *FEMS Microbiol Lett* *70*, 31-36.

Forte, E., Borisov, V.B., Falabella, M., Colaco, H.G., Tinajero-Trejo, M., Poole, R.K., Vicente, J.B., Sarti, P., and Giuffrè, A. (2016). The Terminal Oxidase Cytochrome bd Promotes Sulfide-resistant Bacterial Respiration and Growth. *Scientific Reports* *6*, 23788.

Garg, S., Vitvitsky, V., Gendelman, H.E., and Banerjee, R. (2006). Monocyte differentiation, activation, and mycobacterial killing are linked to transsulfuration-dependent redox metabolism. *J Biol Chem* *281*, 38712-38720.

- Greiner, R., Palinkas, Z., Basell, K., Becher, D., Antelmann, H., Nagy, P., and Dick, T.P. (2013). Polysulfides link H₂S to protein thiol oxidation. *Antioxid Redox Signal* *19*, 1749-1765.
- Hellmich, M.R., Coletta, C., Chao, C., and Szabo, C. (2015). The therapeutic potential of cystathionine β -synthetase/hydrogen sulfide inhibition in cancer. *Antioxid Redox Signal* *22*, 424-448.
- Holyoake, L.V., Hunt, S., Sanguinetti, G., Cook, G.M., Howard, M.J., Rowe, M.L., Poole, R.K., and Shepherd, M. (2016). CydDC-mediated reductant export in *Escherichia coli* controls the transcriptional wiring of energy metabolism and combats nitrosative stress. *Biochem J* *473*, 693-701.
- Hughes, M.N., Centelles, M.N., and Moore, K.P. (2009). Making and working with hydrogen sulfide: The chemistry and generation of hydrogen sulfide *in vitro* and its measurement *in vivo*: a review. *Free Radic Biol Med* *47*, 1346-1353.
- Korshunov, S., Imlay, K.R., and Imlay, J.A. (2016). The cytochrome *bd* oxidase of *Escherichia coli* prevents respiratory inhibition by endogenous and exogenous hydrogen sulfide. *Mol Microbiol* *101*, 62-77.
- Lamprecht, D.A., Finin, P.M., Rahman, M.A., Cumming, B.M., Russell, S.L., Jonnala, S.R., Adamson, J.H., and Steyn, A.J. (2016). Turning the respiratory flexibility of *Mycobacterium tuberculosis* against itself. *Nature Communications* *7*, 12393.
- Moosa, A., Lamprecht, D.A., Arora, K., Barry, C.E., 3rd, Boshoff, H.I.M., Ioerger, T.R., Steyn, A.J.C., Mizrahi, V., and Warner, D.F. (2017). Susceptibility of *Mycobacterium tuberculosis* Cytochrome *bd* Oxidase Mutants to Compounds Targeting the Terminal Respiratory Oxidase, Cytochrome *c*. *Antimicrob Agents Chemother* *61*.
- Perry, T.L., Wright, J.M., Hansen, S., Allan, B.M., Baird, P.A., and MacLeod, P.M. (1980). Failure of aminooxyacetic acid therapy in Huntington disease. *Neurology* *30*, 772-775.
- Pittman, M.S., Corker, H., Wu, G., Binet, M.B., Moir, A.J., and Poole, R.K. (2002). Cysteine is exported from the *Escherichia coli* cytoplasm by CydDC, an ATP-binding cassette-type transporter required for cytochrome assembly. *J Biol Chem* *277*, 49841-49849.
- Poole, R.K., Cozens, A.G., and Shepherd, M. (2019). The CydDC family of transporters. *Res Microbiol*. Shi, L., Sohaskey, C.D., Kana, B.D., Dawes, S., North, R.J., Mizrahi, V., and Gennaro, M.L. (2005). Changes in energy metabolism of *Mycobacterium tuberculosis* in mouse lung and under *in vitro* conditions affecting aerobic respiration. *Proceedings of the National Academy of Sciences of the United States of America* *102*, 15629.
- Szabo, C. (2016). Gasotransmitters in cancer: from pathophysiology to experimental therapy. *Nature Reviews. Drug discovery* *15*, 185-203.
- Szabo, C., Coletta, C., Chao, C., Módis, K., Szczesny, B., Papapetropoulos, A., and Hellmich, M.R. (2013). Tumor-derived hydrogen sulfide, produced by cystathionine- β -synthase, stimulates bioenergetics, cell proliferation, and angiogenesis in colon cancer. *Proceedings of the National Academy of Sciences* *110*, 12474-12479.

REVIEWERS' COMMENTS:

Reviewer #3 (Remarks to the Author):

All raised issues were adequately addressed. The manuscript as revised is improved. The reporting summary and source data files seem fine.